# FROM SPARSE TO STRUCTURED: A NEW PARADIGM FOR GRADIENT-BASED PARAMETER-EFFICIENT FINE-TUNING

## ABSTRACT

Large pre-trained models have demonstrated extensive applications across various fields. However, fine-tuning these models for specific downstream tasks demands significant computational resources and storage. One fine-tuning method, gradient-based parameter selection (GPS), focuses on fine-tuning only the parameters with high gradients in each neuron, thereby reducing the number of training parameters. Nevertheless, this approach increases computational resource requirements and storage demands. In this paper, we propose an efficient gradient-based and regularized fine-tuning method (GRFT) that updates the rows or columns of the weight matrix. We theoretically demonstrate that the rows or columns with the highest sum of squared gradients are optimal for updating. This strategy effectively reduces storage overhead and improves the efficiency of parameter selection. Additionally, we incorporate regularization to enhance knowledge transfer from the pre-trained model. GRFT achieves state-of-the-art performance, surpassing existing methods such as GPS, Adapter Tuning, and LoRA. Notably, GRFT requires updating only 1.22% and 0.30% of the total parameters on FGVC and VTAB datasets, respectively, demonstrating its high efficiency and effectiveness.

## 1 INTRODUCTION

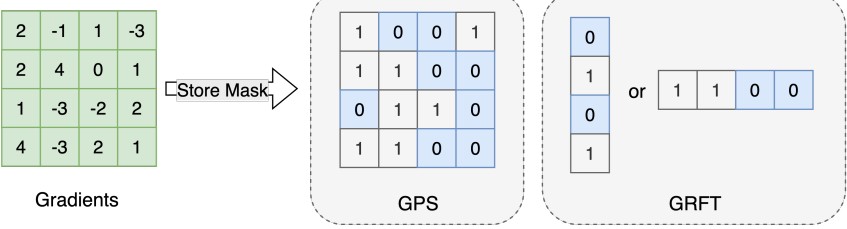

Figure 1: The figure illustrates a comparison between the mask storage formats of the GRFT and GPS schemes. As shown, for a $4 \times 4$ gradient matrix, GPS stores a full $4 \times 4$ sparse mask, whereas GRFT selects either row-wise or column-wise masks, storing only a $4 \times 1$ row index or a $1 \times 4$ column index. This significantly reduces storage overhead.

The applications of large models are expanding rapidly across domains such as natural language processing, computer vision, and scientific research. Multimodal dialogue systems like GPT OpenAI et al. (2024) and LLaMA Touvron et al. (2023) demonstrate their general-purpose adaptability, while in specialized fields, pre-trained models are typically fine-tuned with task-specific datasets to optimize performance for downstream tasks. Despite these successes, fine-tuning large models remains computationally expensive, with excessive memory usage and stringent hardware requirements limiting accessibility for many researchers Li et al. (2024). Moreover, naively reducing parameters often degrades accuracy, making the balance between fine-tuning efficiency and model performance a key research challenge.

To address the high cost of fine-tuning, various parameter-efficient fine-tuning (PEFT) methods have been developed. Approaches such as LoRA Hu et al. (2021), Adapter Tuning Houlsby et al.

(2019b), and Vision Prompt Tuning Jia et al. (2022) introduce small sets of trainable parameters while keeping most weights frozen, thereby improving efficiency. However, these methods often introduce inference latency or compromise the original model structure, reducing expressiveness Li (2024). More recently, Gradient-Based Parameter Selection (GPS) Zhang et al. (2024) directly fine-tunes a subset of the model's own parameters, achieving strong performance. Yet GPS suffers from the need to store large sparse masks, which incurs significant memory overhead and limits hardware efficiency during training and updates.

In our proposed method, we introduce Gradient-based and Regularized Fine-Tuning (GRFT) for parameter-efficient fine-tuning. In particular, rather than selecting sparse parameters, we select entire rows or columns of the weight matrix, meaning that only the indices of these rows or columns need to be stored in the mask. This approach significantly reduces storage costs, while also simplifying the masking mechanism, making it more efficient in terms of memory usage and computational overhead (Figure 1). Additionally, by selecting structured groups of parameters, the method aligns better with modern hardware optimizations, facilitating improved performance during training and inference. Besides, to enhance knowledge transfer from the pre-trained model and improve accuracy, we incorporate an $L_2$ regularization term into the loss function.

To evaluate our method, we conduct experiments and evaluated our method on image classification tasks and text classification tasks. Our proposed method achieves state-of-the-art performance compared to GPS and other PEFT methods while using fewer parameters in certain tasks.

Overall, our contributions are summarized as follows:

- We propose new gradient-based parameter selection frameworks, GRFT, a Gradient-based and Regularized Fine-Tuning method that only trains the parameters associated with large gradients and additional regularization constraints.

- We introduce a novel gradient-based parameter selection method to reduce storage requirements, which fine-tune the entire rows or columns of the parameter matrix. Moreover, a theoretical justification is provided for selecting parameters with larger squared gradients. Additionally, we introduce regularization constraints that limit parameters sizes to be close to those of the pre-trained parameters, thereby facilitating knowledge transfer and enhancing generalization.

- Empirical evaluations over image classification and text classification across ViT models and LLaMA-3 models demonstrate that our method outperforms fine-tuning methods such as GPS and LoRA in terms of accuracy, while not significantly increasing the parameter count.

## 2 RELATED WORKS

### 2.1 PARAMETER-EFFICIENT FINE TUNING

Parameter-efficient fine-tuning is a widely used fine-tuning method in both computer vision and natural language processing, focusing on training parts of the model parameters or fine-tuning additional modules, which has the advantages of lower computational cost and shorter time requirements compared to full fine-tuning. Lately, various existing techniques, including Adapter Tuning Karimi Mahabadi et al. (2021); Lu et al. (2023); Zhang et al. (2023), Prompt Tuning Jia et al. (2022); Zhou et al. (2022); Wang et al. (2023), LoRA Hu et al. (2021) and its variants Qiang et al. (2024); Hayou et al. (2024), are attempting to maintain the model performance while reducing the computation and storage costs. In a recent study, GPS Zhang et al. (2024) fine-tune a few parameters from the pre-trained model while freezing the reminder of the model. The selection of these parameters depends on their individual gradients. The advantage of this method is that it does not introduce additional computational costs and parameters, and it has good adaptability to any agnostic models.

### 2.2 TRANSFER LEARNING AND REGULARIZATION

Fine-tuning is essentially a transfer learning strategy that leverages the knowledge that the model has learned from large-scale datasets, enabling the model to be fine-tuned with a smaller, task-specific dataset. The key advantage of fine-tuning is that it allows the model to converge faster and achieve

better performance with less data compared to training from scratch. However, excessive fine-tuning can lead to catastrophic forgetting. As a result, the model may experience a decline in performance on the tasks Toneva et al. (2018); French (1999). Regularization can improve the generalization ability of models, such as Ridge Regression Hoerl & Kennard (2000). It introduces a regularization term proportional to the square of the magnitude of the model parameters. Additionally, Ridge Regression Hoerl & Kennard (2000) improves the generalization ability of models by preventing overfitting. The $L_2$ regularization provided by Ridge Regression helps reduce this risk by penalizing excessively large weights. In our methods, we add $L_2$ regularization in the training loss function to limit the parameters close to pre-trained parameters, achieveing knowledge transferring and regularization.

## 3 PROPOSED METHOD

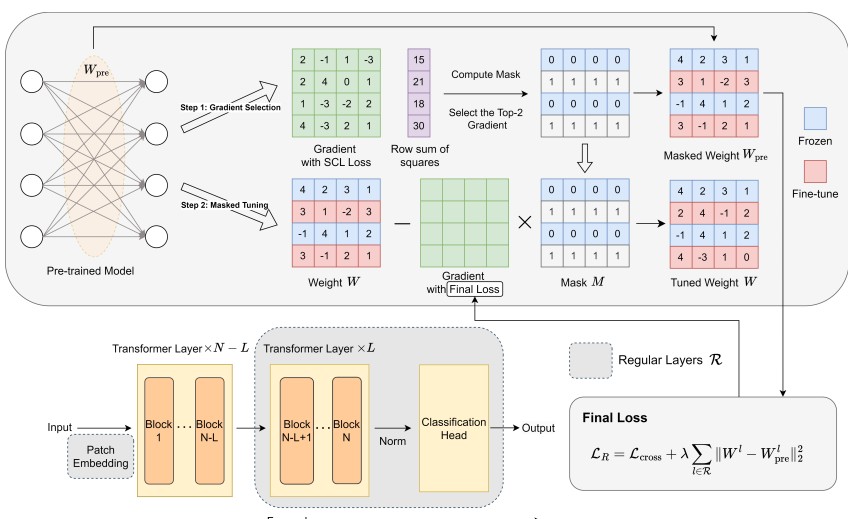

Figure 2: The overall pipeline of our approach. This approach primarily consists of two steps. Step 1: Gradients Selection. Before training, we firstly compute gradients of the pre-trained model with SCL loss and calculate squares sum of the each row. Then,we gain the mask to freezing the parameters. Step 2: Masked Tuning. For pre-trained models, an additional constraint is incorporated alongside the traditional cross-entropy loss function to account for knowledge transfer from the pre-trained models.

In this section, we present the gradient-based fine-tuning approach, a parameter-efficient method that selects and updates parameters with large gradients. The approach consists of two main components: gradient selection and model regularization.

### 3.1 GRADIENT SELECTION

In GPS Zhang et al. (2024), the key step is gradient-based selection, where the top gradients from each neuron are chosen to ensure every neuron contributes to downstream tasks. However, this design poses hardware challenges: gradients must still be computed for the entire weight matrix, with irrelevant parts masked during updates. Since the mask has the same dimensions as the weight matrix, this also results in considerable storage overhead.

**The Principle of Gradient Selection** To address this issue, we adopt a row-wise selection strategy. In the strategy, for the weight matrix $W$, we only select the row parameters with large gradient. Compared to sparse matrices, selecting gradient parameters row by row imposes lower hardware demands and is more beneficial for subsequent applications and practical implementations. Furthermore, the storage cost is reduced since the mask only needs to store the indices of the rows, rather than the entire mask. Specifically, for a weight matrix $W \in \mathbb{R}^{m \times n}$ and its corresponding mask $M$, the GPS approach stores a full mask $M \in \mathbb{R}^{m \times n}$, which is a sparse matrix where each row contains

$K$ ones and the remaining entries are zeros. In contrast, the row-wise or column-wise parameter selection methods only store the indices of the selected rows or columns, resulting in $M \in \mathbb{R}^K$. During the selection process, we typically have $K \ll \min(m, n)$, which leads to a substantial reduction in storage cost.

At the time step $t$ with the model parameters $\theta^t$, we consider $\mathcal{L}$ to be the loss function. The first-order Taylor expansion is

$$\mathcal{L}(\theta^{t+1}) \approx \mathcal{L}(\theta^t) + \langle \nabla \mathcal{L}(\theta^t), \theta^{t+1} - \theta^t \rangle. \tag{1}$$

At the same time, if we assume that the learning rate is $\alpha_t$ at time step $t$, it is evident that for gradient descent optimization, there is $\theta^{t+1} - \theta^t = -\alpha_t \nabla \mathcal{L}(\theta^t) \odot M$. Here $\odot$ is element-wise multiplication and $M$ is the gradient mask to freeze the parameters. Put it in the above Eq. equation 1:

$$\Delta \mathcal{L} = \mathcal{L}(\theta^{t+1}) - \mathcal{L}(\theta^t) \approx -\langle \nabla \mathcal{L}(\theta^t), \alpha_t \nabla \mathcal{L}(\theta^t) \odot M \rangle.$$

We determine $M$ to facilitate a larger decrease in the loss during each iteration:

$$M = \operatorname{argmax}_M \langle \nabla \mathcal{L}(\theta^t), \nabla \mathcal{L}(\theta^t) \odot M \rangle \tag{2}$$

$$= \operatorname{argmax}_M \langle \nabla \mathcal{L}(\theta^t) \odot M, \nabla \mathcal{L}(\theta^t) \odot M \rangle. \tag{3}$$

Note that since $M$ is a binary matrix with values of 0 or 1, when $\nabla \mathcal{L}(\theta^t) = [\nabla \mathcal{L}(\theta^t)_1, \nabla \mathcal{L}(\theta^t)_2, \cdots, \nabla \mathcal{L}(\theta^t)_d] \in \mathbb{R}^d$ is a one-dimensional vector and the mask $M = [M_1, M_2, \cdots, M_d] \in \mathbb{R}^d$, performing the element-wise multiplication with $M$ results in zeros at the corresponding positions. We have

$$\langle \nabla \mathcal{L}(\theta^t), \nabla \mathcal{L}(\theta^t) \odot M \rangle = \sum_{i=1}^d \nabla \mathcal{L}(\theta^t)_i \cdot (\nabla \mathcal{L}(\theta^t)_i \cdot M_i) = \sum_{i=1}^d (\nabla \mathcal{L}(\theta^t)_i \cdot M_i)^2$$

$$= \langle \nabla \mathcal{L}(\theta^t) \odot M, \nabla \mathcal{L}(\theta^t) \odot M \rangle,$$

Therefore, when computing Eq. equation 2, the values at those positions in the final result will also be zero. This leads to the conclusion that Eq. equation 3 holds.

**The Computation of Gradient Mask** Accordingly, we can infer that if $M$ is sparse, the Eq. equation 3 implies a preference for retaining the largest gradients in the entire weight matrix. Therefore, it is reasonable for GPS Zhang et al. (2024) to select based on neurons, retaining the largest value in each row of the gradient matrix. However, the sparse gradient computation has vital hardware requirements. And the mask is stored as the same size of weight matrix, making it a high storage cost. Therefore, our proposed approach is to select the entire row or column, which is theoretically justified to select the rows with the largest squared sums of the gradients. Given the pretrained parameter $W_{\mathrm{pre}}$, we obtain the optimal fine-tuned parameter based on with gradient $\nabla \mathcal{L}^{\mathrm{scl}}(W_{\mathrm{pre}})$ in the following optimization objective,

$$\min_M \| \nabla \mathcal{L}^{\mathrm{scl}}(W_{\mathrm{pre}}) - \nabla \mathcal{L}^{\mathrm{scl}}(W_{\mathrm{pre}}) \odot M \|_2^2 \qquad s.t. \| M_{,j} \|_0 \le k, \quad \forall j \in [n].$$

Note that, to prevent the effects of the randomly initialized classification head on fine-tuning, the SCL loss $\mathcal{L}^{\mathrm{scl}}$ Khosla et al. (2020); Zhang et al. (2024) is used to calculate the gradients.

Based on the conclusion in Eq. equation 3, we compute the sum of the squared gradient of each row. Let the squared sum of the $i_{\mathrm{th}}$ row be defined as:

$$S_i = \sum_{j=1}^n h_{ij}^2, \quad \text{for } i = 1, 2, \ldots, m, \tag{4}$$

where $h_{ij}$ is the element of $H \equiv \nabla \mathcal{L}^{\mathrm{scl}}(W)$ in the $i_{\mathrm{th}}$ row and $j_{\mathrm{th}}$ column. We then select the indices corresponding to the top-$k$ largest values in $\{S_1, S_2, \ldots, S_m\}$, denoted the selected index set as $\mathcal{T}$. A mask $M$ is constructed as follows:

$$M_{ij} = \begin{cases} 1, & \text{if } i \in \mathcal{T}, \\ 0, & \text{otherwise.} \end{cases} \tag{5}$$

Therefore, the mask determines the parameters we select. In the backpropagation update process, only the selected parameters are updated. For the $t$ training epoch, the $l_{\mathrm{th}}$ layer of the model has the following:

$$W_{t+1}^l - W_t^l = -\eta_t \nabla \mathcal{L}(W_t^l) \odot M^l, \tag{6}$$

where $W_t^l$ means the parameters of $l$-th layer at $t$ step and $M^l$ is the mask of $l$-th layer.

---

**Algorithm 1** Gradient-based and Regularized Fine-tuning (GRFT)

---

**Input:** A layer weight matrix $W \in \mathbb{R}^{m \times n}$. Epochs $N$, learning rate $\eta_t$, decay rates $\beta_1$, $\beta_2$, select row number $k$, scale hyperparameter $\lambda$, regular module set $\mathcal{R}$.

1: **Computing mask** $M$
2:     $H \leftarrow \nabla \mathcal{L}^{\text{scl}}(W_{\text{pre}}) \in \mathbb{R}^{m \times n}$
3:     $S_i \leftarrow h_{i1}^2 + h_{i2}^2 + \cdots + h_{in}^2, i = 1, 2, \cdots, m$
4:     Sort $\mathcal{S} = \{S_1, S_2, \ldots, S_m\}$
5:     Obtain the mask $M$ based on Eq. equation 5
6: **Training**
7: **for** $t = 1$ to $N$ **do**
8:     $\mathcal{L}_R = \mathcal{L}_{\text{cross}} + \lambda \sum_{l \in \mathcal{R}} \|W_t^l - W_0^l\|_2^2$
9:     **UPDATE** $(\hat{g}_t)$ **by Adam**:
10:     $\hat{g}_t = \nabla \mathcal{L}_R(W_t) \odot M$
11:     $m_t \leftarrow \beta_1 m_{t-1} + (1 - \beta_1)\hat{g}_t$
12:     $v_t \leftarrow \beta_2 v_{t-1} + (1 - \beta_2)\hat{g}_t^2$
13:     $\hat{m}_t \leftarrow m_t/(1 - \beta_1^t)$
14:     $\hat{v}_t \leftarrow v_t/(1 - \beta_2^t)$
15:     $W_t \leftarrow W_{t-1} - \eta_t \hat{m}_t / \sqrt{\hat{v}_t + \epsilon}$
16:     $t \leftarrow t + 1$
17: **end for**

---

## 3.2 Model Regularization

The essence of fine-tuning large models is to transfer the knowledge acquired during pre-training to downstream tasks. Trained on massive datasets, pre-trained models capture general features and representations that benefit diverse applications. Effective fine-tuning must preserve this knowledge while adapting it to new contexts.

A major challenge in fine-tuning is avoiding catastrophic forgetting Toneva et al. (2018); French (1999), where over-adaptation to a new task causes the model to lose generalizable knowledge from pre-training. To address this, regularization techniques are often introduced during the fine-tuning process. We employ an $L_2$ norm constraint in the loss function to restrict parameter updates during fine-tuning, thereby facilitating the transfer of pre-trained knowledge to downstream tasks. For the classification head, whose parameters are initialized within a small interval around zero due to the large input dimension He et al. (2015), the $L_2$ norm further reduces parameter complexity, helping to prevent overfitting and improve generalization. The overall objective function is

$$\mathcal{L}_R = \mathcal{L}_{\text{cross}} + \lambda \|W - W_{\text{pre}}\|_2^2. \tag{7}$$

where $\mathcal{L}_R$ is the final loss function, $\mathcal{L}_{\text{cross}}$ is the cross-entropy loss function during the training process, $\lambda$ is the regularization parameter for the $L_2$ norm. However, since the parameters of the last few layers of the model have a significant impact on the training results during fine-tuning Zhang et al. (2023), the constraint primarily targets these layers. The final loss function includes both the original loss function and the modified regularization function, specifically as follows:

$$\mathcal{L}_R = \mathcal{L}_{\text{cross}} + \lambda \sum_{l \in \mathcal{R}} \|W^l - W_{\text{pre}}^l\|_2^2, \tag{8}$$

where $\mathcal{R}$ is the regular layers set consisting of the last $L$ layers, patch embedding, and classification head.

**Gradient-based and Regularized Fine-tuning Algorithm** We present the proposed method in Algorithm 1, which consists of two main parts: Mask Computation and Training. Specifically, before training begins, we first compute the model gradients under the SCL loss $\nabla \mathcal{L}^{\text{scl}}(W_{\text{pre}})$. Subsequently, we calculate the sum of squares for each row or column of the gradient $S_i$ and select the top $k$ rows or columns with the largest sums. Based on Eq. equation 5, we compute our mask $M$. During the training phase, for the standard loss function (such as cross-entropy $\mathcal{L}_{\text{cross}}$), we add an $L_2$ regularization term, thereby obtaining the final loss function $\mathcal{L}_R$ as defined in Eq. equation 8. In this process, we employ the Adam Kingma & Ba (2014) optimizer, where we set $\hat{g}$ in Adam to be the masked gradient and then the standard Adam update procedure is applied to iteratively update the weight matrix until convergence.

| Dataset | CUB-2011 | NABirds | Oxford Flowers | Stan.Dogs | Stan.Cars | Mean Acc. | Params.(%) |
|---|---|---|---|---|---|---|---|
| Full | 87.3 | 82.7 | 98.8 | 89.4 | 84.5 | 89.44 | 100.00 |
| Linear | 85.3 | 75.9 | 97.9 | 86.2 | 51.3 | 79.32 | 0.21 |
| Bias Zaken et al. (2022) | 88.4 | 84.2 | 98.8 | 91.2 | 79.4 | 88.40 | 0.33 |
| Adapter Houlsby et al. (2019a) | 87.1 | 84.3 | 98.5 | 89.8 | 68.6 | 85.66 | 0.48 |
| LoRA Hu et al. (2021) | 85.6 | 79.8 | 98.9 | 87.6 | 72.0 | 84.78 | 0.90 |
| VPT-Shallow Jia et al. (2022) | 86.7 | 78.8 | 98.4 | 90.7 | 68.7 | 84.66 | 0.29 |
| VPT-Deep Jia et al. (2022) | 88.5 | 84.2 | 99.0 | 90.2 | 83.6 | 89.10 | 0.99 |
| SSF Lian et al. (2023) | 89.5 | 85.7 | 99.6 | 89.6 | 89.2 | 90.72 | 0.45 |
| SPT-AdapterHe et al. (2023) | 89.1 | 83.3 | 99.2 | 91.1 | 86.2 | 89.78 | 0.47 |
| SPT-LoRA He et al. (2023) | 88.6 | 83.4 | 99.5 | **91.4** | 87.3 | 90.04 | 0.60 |
| GPS* Zhang et al. (2024) | 89.6 | 86.8 | 99.7 | 88.9 | 90.4 | 91.06 | 1.07 |
| GRFT (**ours**) | **90.1** | **87.0** | **99.7** | 89.1 | **90.8** | **91.33** | 1.22 |

Table 1: Comparisons results on FGVC with ViT-B/16 models pre-trained on ImageNet-21K.

## 4 EXPERIMENTS

### 4.1 IMPLEMENTATION

In image classification tasks, the model we implement is vit-base-patch16-224-in21k Dosovitskiy et al. (2020). The model uses $16 \times 16$ image patches as inputs and is pre-trained on ImageNet-21k Deng et al. (2009) at resolution $224 \times 224$. By pretraining on the dataset, the model learns the internal representation of the images, which can be used in downstream tasks to extract features. We use the Adam Kingma & Ba (2014) optimizer and apply a cosine learning rate decay for fine-tuning. Each downstream task is trained for 100 iterations, with an additional 10 warm-up epochs for the learning rate before the training iterations begin. In text classification tasks, we implement Llama3.2 1B model Grattafiori & etc. (2024) and fine-tuning in CoLA, MRPC and RTE datasets of GLUE benchmark Wang et al. (2019). We added a linear classification head module to the model to perform our classification task. The optimizer we use is AdamW Loshchilov & Hutter (2019).

### 4.2 DATASETS

**FGVC (Fine-Grained Visual Classification)**: FGVC is a subset of image classification tasks which mainly deal with distinguishing between visually similar objects within a category. FGVC datasets include: Stanford Dogs Khosla et al. (2011), Stanford Cars Krause et al. (2013), Nabirds Van Horn et al. (2015), CUB_200_2011 Wah et al. (2011), Oxfordflower 102 Nilsback & Zisserman (2008).

**VTAB (Visual Task Adaptation Benchmark)**: a benchmark designed to evaluate the performance of transfer learning techniques in visual tasks, testing the performance of models trained on one set of tasks generalizing to a wide variety of other visual tasks. VTAB includes 19 different datasets, covering various visual domains. Natural: includes tasks like CIFAR-100 Krizhevsky et al. (2009) and Caltech101 Fei-Fei et al. (2004). Specialized: includes tasks like Patch Camelyon Veeling et al. (2018) and Resisc45 Cheng et al. (2017). Structured: includes tasks like DMLab Zhai et al. (2020) and Clevr Johnson et al. (2017).

**GLUE (General Language Understanding Evaluation)** Wang et al. (2019): a benchmark dataset designed to measure the capabilities of models in natural language understanding (NLU). It consists of various subtasks, including text classification, sentence similarity evaluation, and natural language inference (NLI), among others. GLUE is primarily used to assess the performance of pre-trained language models.

### 4.3 EXPERIMENTAL RESULTS

We compare with different fine-tuning methods, including full fine-tuning, linear and bias Zaken et al. (2022), Adapter Houlsby et al. (2019a), LoRA Hu et al. (2021), VPT Jia et al. (2022), SSF Lian et al. (2023), SPT He et al. (2023) and GPS Zhang et al. (2024). Except for GPS, the results of the other methods follow the results in the GPS paper, while the GPS results are reproduced by ourselves, marked as GPS*. Our results are shown in Table 1 and Table 2.

| Method / Dataset | Natural | | | | | | | Specialized | | | | Structured | | | | | | | | VTAB | |
|---|---|---|---|---|---|---|---|---|---|---|---|---|---|---|---|---|---|---|---|---|---|
| | CIFAR-100 | Caltech101 | DTD | Flowers102 | Pets | SVHN | Sun397 | Patch Camelyon | EuroSAT | Resisc45 | Retinopathy | Clevr/count | Clevr/distance | DMLab | KITTI/distance | dSprites/loc | dSprites/ori | SmallNORB/azi | SmallNORB/ele | Mean Acc. | Mean Params. (%) |
| Full | 68.9 | 87.7 | 64.3 | 97.2 | 86.9 | 87.4 | 38.8 | 79.7 | 95.7 | 84.2 | 73.9 | 56.3 | 58.6 | 41.7 | 65.5 | 57.5 | 46.7 | 25.7 | 29.1 | 65.57 | 100.00 |
| Linear | 63.4 | 85.0 | 64.3 | 97.0 | 86.3 | 36.6 | 51.0 | 78.5 | 87.5 | 68.6 | 74.0 | 34.3 | 30.6 | 33.2 | 55.4 | 12.5 | 20.0 | 9.6 | 19.2 | 53.00 | 0.05 |
| Bias Zaken et al. (2022) | 72.8 | 87.0 | 59.2 | 97.5 | 85.3 | 59.9 | 51.4 | 78.7 | 91.6 | 72.9 | 69.8 | 61.5 | 55.6 | 32.4 | 55.9 | 66.6 | 40.0 | 15.7 | 25.1 | 62.05 | 0.16 |
| Adapter Houlsby et al. (2019a) | 74.1 | 86.1 | 63.2 | 97.7 | 87.0 | 34.6 | 50.8 | 76.3 | 88.0 | 73.1 | 70.5 | 45.7 | 37.4 | 31.2 | 53.2 | 30.3 | 25.4 | 13.8 | 22.1 | 55.82 | 0.31 |
| LoRA Hu et al. (2021) | 68.1 | 91.4 | 69.8 | 99.0 | 90.5 | 86.4 | 53.1 | 85.1 | 95.8 | 84.7 | 74.2 | 83.0 | 66.9 | 50.4 | 81.4 | 80.2 | 46.6 | 32.2 | 41.1 | 72.63 | 0.90 |
| VPT-Shallow Jia et al. (2022) | 77.7 | 86.9 | 62.6 | 97.5 | 87.3 | 74.5 | 51.2 | 78.2 | 92.0 | 75.6 | 72.9 | 50.5 | 58.6 | 40.5 | 67.1 | 68.7 | 36.1 | 20.2 | 34.1 | 64.85 | 0.13 |
| VPT-Deep Jia et al. (2022) | 78.8 | 90.8 | 65.8 | 98.0 | 88.3 | 78.1 | 49.6 | 81.8 | 96.1 | 83.4 | 68.4 | 68.5 | 60.0 | 46.5 | 72.8 | 73.6 | 47.9 | 32.9 | 37.8 | 69.43 | 0.70 |
| SSF Lian et al. (2023) | 69.0 | 92.6 | 75.1 | 99.4 | 91.8 | 90.2 | 52.9 | 87.4 | 95.9 | 87.4 | 75.5 | 75.9 | 62.3 | 53.3 | 80.6 | 77.3 | 54.9 | 29.5 | 37.9 | 73.10 | 0.28 |
| SPT-ADAPTER He et al. (2023) | 72.9 | 93.2 | 72.5 | 99.3 | 91.4 | 88.8 | 55.8 | 86.2 | 96.1 | 85.5 | 75.5 | 83.0 | 68.0 | 51.9 | 81.2 | 51.9 | 31.7 | 41.2 | 61.4 | 73.03 | 0.44 |
| SPT-LoRA He et al. (2023) | 73.5 | 93.3 | 72.5 | 99.3 | 91.5 | 87.9 | 55.5 | 85.7 | 96.2 | 85.9 | 75.9 | 84.4 | 67.6 | 52.5 | 82.0 | 81.0 | 51.1 | 30.2 | 41.3 | 74.07 | 0.63 |
| GPS* Zhang et al. (2024) | 68.7 | 93.6 | 72.6 | 99.3 | 90.0 | 90.1 | 52.4 | 87.0 | 95.9 | 86.5 | 76.1 | 78.9 | 62.2 | 54.7 | 79.7 | 80.8 | 54.9 | 30.7 | 44.6 | 73.61 | 0.24 |
| GRFT(ours) | 69.5 | 93.6 | 75.9 | 99.5 | 91.4 | 91.2 | 52.2 | 88.2 | 96.0 | 86.5 | 76.3 | 81.4 | 62.3 | 55.1 | 80.9 | 81.9 | 55.8 | 32.0 | 43.6 | 74.38 | 0.30 |

Table 2: Comparisons results on VTAB-1k with ViT-B/16 models pre-trained on ImageNet-21K.

In Table 1, The average accuracy of GRFT is the highest, reaching 91.33%, indicating that it has better generalization ability when handling different datasets. In contrast, the average accuracies of the other models range from 84.66% (VPT-Shallow) to 91.06% (GPS*), all of which are lower than GRFT. In terms of the number of parameters, GRFT has 1.22% of the total parameters, which is considered above average among all methods. This suggests that GRFT achieves a high accuracy in the large datasets while increasing the number of parameters to be updated. Some other methods, although having fewer parameters, also show relatively lower accuracy. From this, it can be concluded that GRFT performs exceptionally well in the FGVC experiment.

In Table 2, The GRFT achieves an impressive mean accuracy of 74.38%, which is the highest among all the methods listed in the table. For instance, GRFT outperforms GPS*, which has a mean accuracy of 73.89%, and SPT-LoRA, which has a mean accuracy of 74.07%. The GRFT operates with a mean parameter percentage of 0.30%, which is relatively low compared to some other methods. This suggests that GRFT can achieve high accuracy, making it more efficient in terms of computational resources. In summary, the GRFT stands out for its high accuracy and parameter efficiency across a wide range of datasets. Its ability to achieve superior results with fewer parameters makes it a promising candidate for applications where computational efficiency and model compactness are critical.

| Dataset | CoLA | MRPC | RTE | Mean Acc. | Params.(%) |
|---|---|---|---|---|---|
| Full | 0.8428 | 0.8603 | 0.8087 | 0.8373 | 100.00 |
| LoRA Hu et al. (2021) | 0.8562 | 0.8554 | 0.8159 | 0.8425 | 0.19 |
| GRFT(ours) | 0.8495 | 0.8554 | 0.8484 | **0.8511** | **0.08** |

Table 3: Comparisons results on sub-tasks of the GLUE with LLaMA3-1B models on a single GPU.

We conducted experiments on Llama3, with the results presented in Table 3. Due to the high storage requirements of GPS, it was not feasible to run it on Llama3. Therefore, we compared our method with full fine-tuning and LoRA. The results clearly demonstrate that our approach exhibits strong adaptability across different models, ensuring generalization and enabling efficient fine-tuning across various model architectures. Additionally, our method significantly reduces computational and storage overhead while maintaining high performance, making it more practical for real-world applications with resource constraints.

To analyze storage efficiency, we conducted a comparative study on the FGVC dataset. We examined the time interval from the onset of training to the point of convergence, during which the trajectory of GPU memory allocation over time is depicted in Fig. 3.

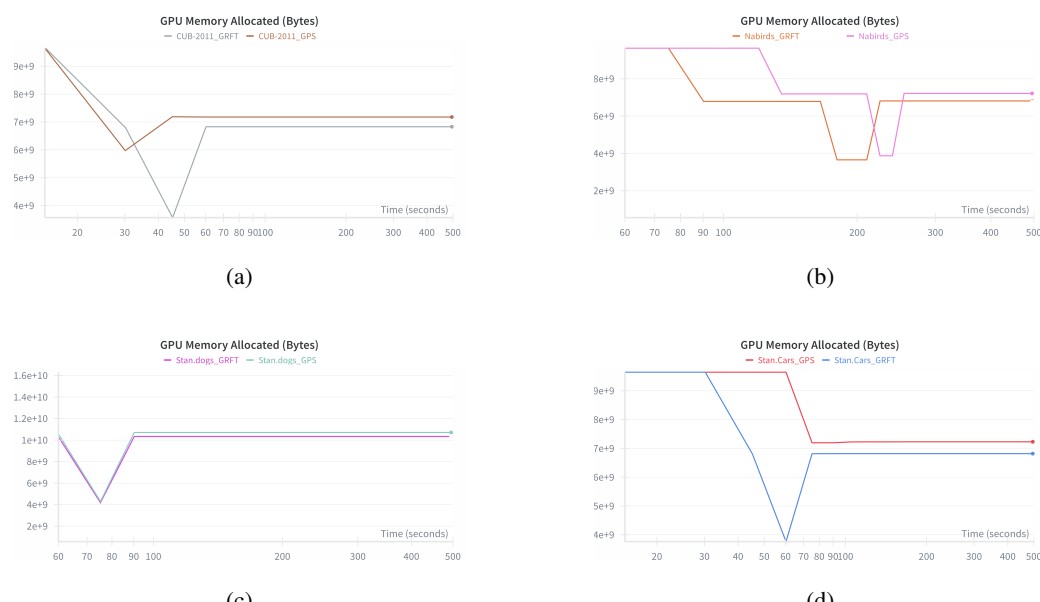

Figure 3: The GPU memory usage curve

## 4.4 ABLATION STUDY

### 4.4.1 SELECTED ROWS OR COLUMNS NUMBER

We select the top $k$ rows or columns of each gradient matrix as the trainable parameters, with $k$ ranging from 1 to 30, and conduct experiments across multiple tasks. It can be observed that having more trainable parameters does not necessarily lead to better performance; instead, each dataset exhibits a performance peak. Furthermore, on larger datasets, adding trainable parameters significantly enhances accuracy. By controlling the number of trainable parameters, it is possible to achieve optimal results across different datasets. The Rows number results are shown in Fig. 4 (a).

### 4.4.2 SELECTED METHODS: SPARSE, ROW, OR COLUMN?

Our approach incorporates two selection methods: one based on selecting rows and the other on selecting columns. To investigate the differences among these two methods and sparse selection, we conducted an ablation experiment on the FGVC dataset, and the performance results are presented in Table 4. Our findings reveal that while both rows and columns selection types exhibit comparable overall performance, they yield different results depending on the specific characteristics of the data. And the row/column selection outperforms the sparse selection scheme. This suggests that the choice of rows/columns selection method may have varying impacts on model performance, influenced by the structure and nature of the dataset.

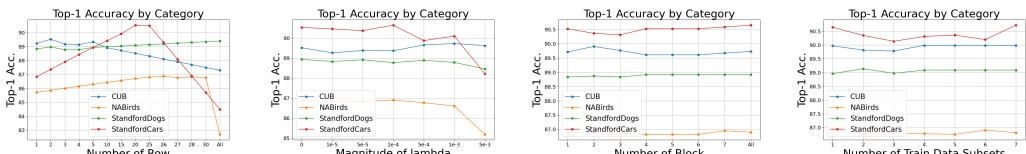

Figure 4: Impacts of different setting. From left to right: (a) Impacts of different numbers of selected rows on performance. (b) Impacts of different Magnitude of lambda in $L_2$ norm on performance. (c) Impacts of different numbers of selected block on performance. (d) Impacts of different numbers of train data subsets on performance.

| Dataset | CUB-2011 | NABirds | Oxford Flowers | Stan.Dogs | Stan.Cars | Mean Acc. |
|---|---|---|---|---|---|---|
| GPS* Zhang et al. (2024) | 89.6 | 86.8 | 99.7 | 88.9 | 90.4 | 91.06 |
| GRFT-Sparse | 89.7 | 86.8 | 99.7 | 88.9 | 90.6 | 91.13 |
| GRFT-Row | 90.0 | 87.0 | 99.7 | 89.1 | 90.7 | 91.29 |
| GRFT-Column | 90.1 | 86.9 | 99.7 | 88.8 | 90.8 | 91.27 |

Table 4: Comparisons results on FGVC in different selected methods.

### 4.4.3 Types of Regularization Norms

The aims of regularization are two aspects. Firstly, it ensures that the model parameters are updated in the vicinity of the pre-trained model's weights, facilitating the transfer of knowledge from the pre-trained model. Secondly, it enhances the model's generalization ability. There are various choices of regularization norms, and here we focus on comparing the performance differences between $L_1$ and $L_2$ norms under the same parameter settings. The experimental results are shown in Table 5. From the results, we observe that $L_2$ regularization generally outperforms $L_1$ regularization.

| Dataset | CUB-2011 | NABirds | Oxford Flowers | Stan.Dogs | Stan.Cars | Mean Acc. |
|---|---|---|---|---|---|---|
| Without Norm | 89.5 | 86.9 | 99.6 | 88.9 | 90.5 | 91.10 |
| $L_1$ Norm | 89.8 | 85.8 | 99.6 | 88.8 | 88.7 | 90.54 |
| $L_2$ Norm | 89.9 | 86.9 | 99.7 | 89.1 | 90.6 | 91.25 |

Table 5: Comparisons results on FGVC in different norms.

### 4.4.4 Magnitude of Regularization Parameter

After determining the selection and regularization methods, we tested different values of $\lambda$ ranging from $1 \times 10^{-8}$ to $1 \times 10^{-3}$ across tasks. As shown in Fig. 4(b), each dataset achieves its best performance at an optimal $\lambda$.

### 4.4.5 Number of Regular Blocks

Once $\lambda$ was determined, we further tested the number of regular blocks, ranging from 1 to 8, and selected the best performing configuration as the optimal result for the dataset under our proposed approach Fig.4 (c).

### 4.4.6 Data Processing

We preprocess the training data by randomly splitting it into $n$ subsets and selecting the one with minimal loss to compute the mask and determine trainable parameters. The hyperparameter $n$ controls the number of subsets (from 1 to 7, where $n = 1$ means no partitioning). Results under different $n$ are shown in Fig. 4(d).

## 5 Conclusion

In this paper, we proposed Gradient-based and Regularized Fine-Tuning (GRFT), an innovative parameter-efficient fine-tuning method. GRFT selects structured groups of parameters corresponding to rows or columns with the largest sum of squared gradients to update, while incorporating $L_2$ regularization to mitigate the challenges of computational and storage inefficiency and to preserve knowledge when adapting large pre-trained models to downstream tasks. GRFT demonstrates significant improvements in average accuracy on FGVC and VTAB, outperforming existing parameter-efficient fine-tuning (PEFT) methods. Future work can explore the integration of GRFT with continual learning techniques to enable lifelong adaptation across evolving tasks without excessive computational cost.

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

# A APPENDIX

## A.1 THE USE OF LARGE LANGUAGE MODELS (LLMS)

In the preparation of this paper, we employed large language models (LLMs) as auxiliary tools to polish the language. Specifically, LLMs were used to refine grammar, improve fluency, and enhance readability of the manuscript. It is important to note that LLMs were not involved in generating the core content, experiments, or results of this work. Their role was limited to linguistic assistance, ensuring the clarity and academic style of the final presentation.

## A.2 BASELINE DESCRIPTION

### A.2.1 GPS

GPS (Gradient-based Parameter Selection) is an innovative Parameter-Efficient Fine-Tuning (PEFT) method designed to address the computational and storage challenges associated with fine-tuning large-scale pretrained models on downstream tasks. Compared to traditional full-parameter fine-tuning approaches, GPS achieves efficient model adaptation by adjusting only a small subset of key parameters in the pretrained model while keeping the remaining parameters frozen. This significantly reduces computational costs and memory consumption.

The core idea of GPS is to select the most critical parameters for a downstream task based on gradient information. Specifically, the method first computes the gradient values of each neuron's input connections, where the magnitude of the gradient reflects the importance of the parameter in the current task. GPS selects parameters with the highest gradient values, as these parameters exhibit the most rapid changes in the loss function and contribute the most to model performance improvement. Additionally, to ensure that the model can adjust to features at different levels, GPS employs a distributed parameter selection strategy—rather than simply selecting the parameters with the highest gradients across the entire network, it selects the top input connections within each neuron. This strategy ensures a more balanced parameter distribution across different layers of the model, allowing for better adaptation to the feature requirements of downstream tasks.

GPS offers several significant advantages. First, it does not introduce any additional parameters, thereby avoiding increased computational overhead during both training and inference. Second, GPS is model-agnostic and can be applied to various architectures, such as Transformers and CNNs, without requiring modifications to the model structure. Furthermore, GPS dynamically selects parameters based on the specific needs of each downstream task, leading to improved adaptability and overall performance.

### A.2.2 LoRA

LoRA (Low-Rank Adaptation) is an efficient parameter adaptation method specifically designed for fine-tuning large-scale pre-trained language models. It adapts to downstream tasks by injecting trainable low-rank factorized matrices into each layer of the Transformer architecture, while keeping the pre-trained weights frozen. This significantly reduces the number of trainable parameters required for the downstream task. The core idea of LoRA is based on the assumption that the weight changes during model adaptation have low "intrinsic rank," meaning they can be approximated by low-rank matrices. This approach allows LoRA to substantially reduce computational and storage costs while maintaining model performance, and it does not introduce additional inference latency during deployment. LoRA has shown outstanding performance across multiple natural language processing tasks, being competitive with full-parameter fine-tuning in terms of model quality, while significantly reducing the number of trainable parameters and GPU memory requirements.

### A.3 THE DIFFERENCES BETWEEN GPS AND GRFT

#### A.3.1 PARAMETER SELECTION STRATEGY

The differences of strategy is shown as Fig.5.

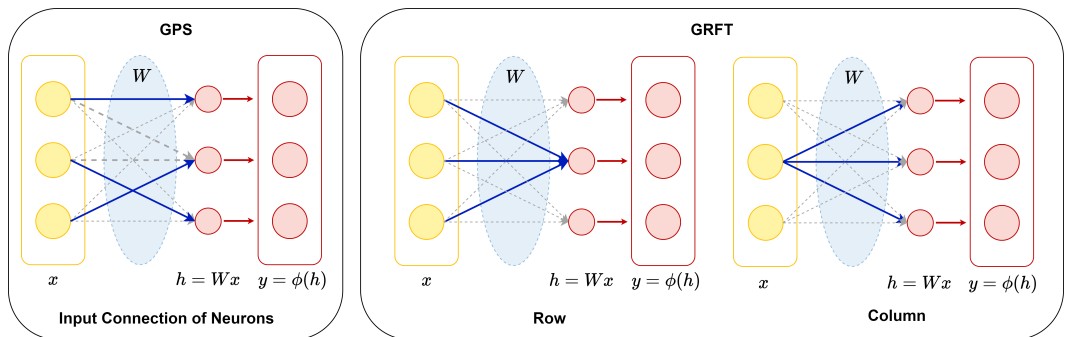

Figure 5: The differences selection methods between GPS and GRFT. The illustration depicts the parameter selection when $K = 1$ for both methods, which marked in blue. Specifically, GPS selects one connection per neuron, while GRFT selects an entire row or column. The selection pattern of GPS is sparse, meaning that each row of the parameter matrix updates only a single element. In contrast, GRFT updates an entire row of the weight matrix at once.

- **GPS:** GPS selects parameters for fine-tuning by computing the gradient values of each neuron's input connections. For each neuron, it selects the top-$K$ parameters with the

highest gradient values. This approach ensures that the selected parameters undergo the most rapid changes in the loss function, enabling the model to quickly adapt during fine-tuning. GPS relies on a sparse matrix-based parameter selection strategy, requiring a mask matrix of the same dimensions as the weight matrix, which increases storage costs.

- **GRFT:** GRFT selects entire rows or columns of the weight matrix for fine-tuning instead of individual parameters. Specifically, it selects the top-$K$ rows or columns with the highest gradient squared sum. This method reduces storage costs as it only requires storing the indices of the selected rows or columns, rather than a full mask matrix. GRFT's parameter selection strategy is more efficient in hardware implementation since it avoids the computational complexity associated with sparse matrix operations.

To analyze the differences between row selection and column selection, we have conducted a simple preliminary analysis, as illustrated in the Fig. 5. For a weight matrix $W$ and its corresponding gradient matrix of the same dimensions, selecting a row corresponds to the parameters that represent all input connections for a specific feature component in the output vector $y$, while selecting a column corresponds to all output connections for a specific feature component in the input vector $x$. We assume that row selection focuses on the complete feature from the previous layer to a specific feature component in the current layer, and during back propagation, it represents the impact of the feature component in the current layer on the previous layer. In contrast, column selection focuses on the influence from a specific feature component in the previous layer to the features in the next layer. The choice of which one to select in practice, along with the related principles, can be left for future research.

### A.3.2 REGULARIZATION STRATEGY

- **GPS:** GPS does not introduce additional regularization strategies. It primarily relies on gradient selection for parameter optimization.

- **GRFT:** GRFT incorporates $L_2$ regularization by adding a regularization term to the loss function. This constrains the updates of fine-tuned parameters to remain close to those of the pre-trained model. The regularization strategy helps prevent excessive parameter adjustments during fine-tuning, preserves the knowledge acquired in the pre-training phase, and mitigates catastrophic forgetting.

### A.3.3 STORAGE

- **GPS:** GPS requires storing a mask matrix of the same dimensions as the weight matrix, leading to increased storage costs.

- **GRFT:** GRFT only requires storing the indices of selected rows or columns, significantly reducing storage costs.

### A.4 EXPERIMENT DETAILS

In this section, we present the relevant experimental parameter settings for our image classification tasks on FGVC and VTAB. The tables below include several parameters associated with the methods discussed in the paper. Their specific meanings are as follows: Data Subsets Number denotes the number of splits in the training dataset during data processing, Regular Parameter refers to the scale parameter in regularization, Regular Layer Number indicates the number of modules added for regularization constraints, with all presenting the all layers in model being added for regularization, and Row/Column Number represents the number of specific rows/columns selected during the fine-tuning process, corresponding to the number of training parameters.

### A.4.1 EXPERIMENTS ON FGVC

We provide a detailed description of the experimental setup and results on the FGVC task. We list the hyperpamameters of the best performance on FGVC in Table 6.

| Dataset | Learning Rate | Batch size | Epoch | Data Subsets Number | Regular Parameter | Regular Layer Number | Row/Column Number |
|---------|------|------|-----|-----|------|-----|-----|
| CUB-2011 | 5e-3 | 32 | 100 | 4 | 1e-3 | 2 | 2 |
| NaBirds | 1e-4 | 32 | 100 | 1 | 1e-4 | 7 | 26 |
| Oxford Flowers | 1e-3 | 32 | 100 | 1 | 1e-4 | 2 | 1 |
| Stan. Dogs | 2e-4 | 64 | 100 | 2 | 5e-5 | 4 | 5 |
| Stan. Cars | 5e-4 | 32 | 100 | 7 | 1e-4 | all | 20 |

Table 6: Hyperparameters on FGVC

### A.4.2 EXPERIMENTS ON VTAB

We present explanations of the experimental setup and results for the VTAB task. We list the hyper-pamameters of the best performance on VTAB in Table 7.

| Dataset | Learning Rate | Batch size | Epoch | Data Subsets Number | Regular Parameter | Regular Layer Number | Row/Column Number |
|---------|------|------|-----|-----|------|-----|-----|
| CIFAR-100 | 2e-3 | 32 | 100 | 1 | 1e-6 | all | 1 |
| Caltech101 | 2e-3 | 16 | 100 | 1 | 1e-8 | 2 | 1 |
| DTD | 2e-3 | 16 | 100 | 1 | 1e-8 | 3 | 1 |
| Flowers102 | 2e-3 | 16 | 100 | 1 | 1e-6 | all | 1 |
| Pets | 3e-3 | 32 | 100 | 1 | 1e-3 | all | 2 |
| SVHN | 5e-3 | 32 | 100 | 1 | 1e-3 | 1 | 4 |
| Sun397 | 2.5e-3 | 16 | 100 | 1 | 1e-6 | 2 | 1 |
| Patch Camelyon | 4e-3 | 32 | 100 | 1 | 1e-7 | all | 2 |
| EuroSAT | 2e-3 | 16 | 100 | 1 | 1e-4 | 3 | 2 |
| Resisc45 | 1.5e-3 | 16 | 100 | 1 | 1e-3 | all | 3 |
| Retinopathy | 2e-3 | 32 | 100 | 1 | 1e-6 | 1 | 1 |
| Clevr/count | 3e-4 | 16 | 100 | 1 | 0 | 0 | 3 |
| Clevr/distance | 2e-3 | 16 | 100 | 1 | 1e-5 | all | 2 |
| DMLab | 1.5e-3 | 16 | 100 | 1 | 0 | 0 | 2 |
| KITTI/distance | 1e-3 | 16 | 100 | 1 | 1e-5 | all | 1 |
| dSprites/loc | 7e-3 | 32 | 100 | 1 | 1e-4 | 5 | 3 |
| dSprites/ori | 5e-4 | 16 | 100 | 1 | 0 | 0 | 2 |
| SmallNORB/azi | 3e-3 | 32 | 100 | 1 | 1e-8 | all | 3 |
| SmallNORB/ele | 3e-4 | 32 | 100 | 1 | 1e-3 | 2 | 2 |

Table 7: Hyperparameters on VTAB

| Dataset | Params. (%) | Row/Column Number | Dataset | Params. (%) | Row/Column Number | Dataset | Params. (%) | Row/Column Number |
|---------|------|------|---------|------|------|---------|------|------|
| CUB-2011 | 0.47 | 2 | Pets | 0.28 | 2 | DMLab | 0.30 | 2 |
| NaBirds | 2.56 | 26 | SVHN | 0.41 | 4 | KITTI/distance | 0.18 | 1 |
| Oxford.Flowers | 0.26 | 1 | Sun397 | 0.53 | 1 | dSprites/loc | 0.34 | 3 |
| Stan.Dogs | 0.58 | 5 | Patch Camelyon | 0.25 | 2 | dSprites/ori | 0.31 | 2 |
| Stan.Cars | 2.22 | 20 | EuroSAT | 0.26 | 2 | SmallNORB/azi | 0.34 | 3 |
| CIFAR-100 | 0.26 | 1 | Resisc45 | 0.37 | 3 | SmallNORB/ele | 0.30 | 2 |
| Caltech101 | 0.26 | 1 | Retinopathy | 0.18 | 1 | CoLA | 0.08 | 3 |
| DTD | 0.22 | 1 | Clevr/count | 0.33 | 3 | MRPC | 0.08 | 3 |
| Flowers102 | 0.26 | 1 | Clevr/distance | 0.26 | 2 | RTE | 0.08 | 3 |

Table 8: The number of learnable parameters across all tasks.

### A.4.3 EXPERIMENTS ON GLUE

For GLUE, we test only three datasets for training to demonstrate the generalizability of our method across different models. Since we used the Llama 3 model and ran it on a single GPU, GPS requires storing a mask of the same size as the model, leading to excessive memory usage that caused the experiment to be unfeasible. Therefore, we compared our method with the full model and LoRA. This comparison further highlights the applicability of GRFT in large models.

### A.4.4 THE NUMBER OF TRAINING PARAMETERS ON DIFFERENT TASKS

For neural networks, our method selects entire rows or columns of parameters, as shown in the Fig.5. For datasets with larger volumes of data, such as Nabirds, we can choose more rows and columns to increase the training parameters, which can improve the model's performance. The Table 8 below shows the proportion of parameters selected in our paper. The datasets include all tasks from FGVC, VTAB and GLUE. The Param represents the proportion of parameters updated using GRFT in a given task relative to the total model parameters. The Row/Column Number indicates the number of selected rows and columns. For most tasks in this paper, we only select no more than five rows/columns.

