# OpenReview forum: "From Sparse to Structured: A New Paradigm for Gradient-Based Parameter-Efficient Fine-Tuning"
_ICLR.cc/2026/Conference — Submitted to ICLR 2026_

### Official Review · Reviewer_KeGi · 2025-10-20

**Soundness:** 2
**Presentation:** 3
**Contribution:** 2
**Rating:** 2
**Confidence:** 4

**Summary:**

The paper introduces GRFT (Gradient-based and Regularized Fine-Tuning), a parameter-efficient fine-tuning approach. Unlike GPS, which selects sparse individual parameters, GRFT updates entire rows or columns of weight matrices that have the largest summed squared gradients. This structured selection reduces storage overhead and improves hardware efficiency compared to GPS. In addition, an L2 regularization term is incorporated to preserve the knowledge learned during pre-training. Experiments on FGVC, VTAB, and GLUE benchmarks demonstrate consistent performance improvements over GPS, LoRA, and Adapter methods.

**Strengths:**

1. The paper is clearly written and well organized. The presentation is smooth and accessible, making the core ideas and technical details easy to understand.
2. The paper shifts gradient-based PEFT from unstructured sparsity to structured row/column selection, which significantly reduces parameter storage and improves computational efficiency.

**Weaknesses:**

1. The novelty is somewhat incremental, mainly extending GPS from element-wise to structured row/column selection. In addition, the proposed regularization contributes only marginal performance gains according to the results presented in Table 5.
2. The experimental results raise some concerns. While GRFT’s storage efficiency is clear, the performance improvement over GPS is less intuitive. Intuitively, GPS performs finer-grained parameter updates, and the paper does not provide sufficient explanation or analysis to clarify why the structured updates in GRFT lead to higher performance.
3. The paper lacks comparisons with several recent and stronger baselines on vision tasks, such as MLAE [1] and GLoRA [2], which limits the assessment of its competitiveness. Meanwhile, the baselines in the LLM experiments are also insufficient, as GPS itself is not included and no comparisons are made with more advanced fine-tuning methods. Moreover, only three datasets from the GLUE benchmark are reported instead of the full suite, which is unusual and raises concerns about possible selective reporting of favorable results.

------

[1] MLAE: Masked LoRA Experts for Visual Parameter-Efficient Fine-Tuning. https://arxiv.org/abs/2405.18897

[2] One-for-All: Generalized LoRA for Parameter-Efficient Fine-tuning. https://arxiv.org/abs/2306.07967

**Questions:**

1. The paper should include a comprehensive comparison of computational efficiency between GRFT, GPS, and LoRA.

---

> ### Author Response · Authors · 2025-11-21
>
> Thank you for your thoughtful and constructive comments. We address each point raised below.
>
> > The novelty is somewhat incremental, mainly extending GPS from element-wise to structured row/column selection. In addition, the proposed regularization contributes only marginal performance gains according to the results presented in Table 5.
>
> > The experimental results raise some concerns. While GRFT’s storage efficiency is clear, the performance improvement over GPS is less intuitive. Intuitively, GPS performs finer-grained parameter updates, and the paper does not provide sufficient explanation or analysis to clarify why the structured updates in GRFT lead to higher performance.
>
> Regarding your concern about novelty, we would like to clarify that the core innovation lies in the structured gradient selection paradigm, which addresses key storage issues that GPS cannot solve. We propose replacing GPS’s element-wise sparse parameter selection with a row/column structured parameter selection method GRFT, which changes the way the mask is stored and effectively addresses the issue of excessively large mask storage caused by sparse selection.
>
> 1. Structured Selection in GRFT vs. GPS: The structured selection in GRFT is fundamentally different from the element-wise sparsity selection in GPS, both theoretically and in implementation. The former can effectively reduce storage and access overhead. We have demonstrated this with GPU memory usage curves on several fgvc datasets in Figure 3.
>
> 2. Theoretical Justification: In our work, we prove that row/column selection is the optimal structured selection approach. In Section 3.1, we provide a theoretical proof. Under the structured constraint, we maximize the change in the loss, which leads to the optimal solution for selecting the rows/columns with the largest gradient squared sums:
>    $$
>    M=\text{argmax}_M \langle\nabla \mathcal{L}(\theta^{t}), \nabla \mathcal{L}(\theta^t) \odot M \rangle
>    =\text{argmax}_M \langle\nabla \mathcal{L}(\theta^{t})\odot M, \nabla \mathcal{L}(\theta^t)\odot M \rangle
>    $$
>
> We acknowledge that the primary contribution of GRFT lies in efficiency rather than significant accuracy improvements. Nevertheless, GRFT achieves accuracy comparable to the main baseline, GPS, while offering substantial storage efficiency. This efficiency advantage is crucial for future optimization and practical applications. For instance, on LLaMA-3 1B, GPS fails to run, whereas GRFT operates normally.
>
> | Dataset | CoLA   | MRPC   | RTE    | Mean Acc. | Params.(%) |
> | ------- | ------ | ------ | ------ | --------- | ---------- |
> | Full    | 0.8428 | 0.8603 | 0.8087 | 0.8373    | 100.00     |
> | LoRA    | 0.8562 | 0.8554 | 0.8159 | 0.8425    | 0.19       |
> | GRFT    | 0.8495 | 0.8554 | 0.8484 | 0.8511    | 0.08       |
>
> The table above presents results for several small GLUE tasks, demonstrating GRFT's versatility in multimodal tasks and large model applications.

---

> ### Author Response · Authors · 2025-11-21
>
> Due to the word limit, I will reply in two parts. Continuing from the previous response:
> > The paper lacks comparisons with several recent and stronger baselines on vision tasks, such as MLAE [1] and GLoRA [2], which limits the assessment of its competitiveness. Meanwhile, the baselines in the LLM experiments are also insufficient, as GPS itself is not included and no comparisons are made with more advanced fine-tuning methods. Moreover, only three datasets from the GLUE benchmark are reported instead of the full suite, which is unusual and raises concerns about possible selective reporting of favorable results.
>
> Thank you for your feedback. We would like to clarify that we did not intentionally select datasets to achieve ideal experimental results. We only chose the three smallest datasets from GLUE for testing. The main reason is that we wanted to demonstrate the generalizability of our approach across modalities by comparing on text-based tasks, and the smaller datasets were sufficient to illustrate this point.
>
> Regarding the comparison with the MLAE and GLoRA baselines, we encountered an issue during our experiments: the average accuracy reported for these two methods on VTAB-1k was inconsistent with the task-wise accuracies shown in their respective tables.
>
> We performed a detailed review of the VTAB-1k tables in these two papers. By calculating the arithmetic average of the accuracy across the 19 tasks, the result did not match the "average" value in the last column of the papers, with discrepancies of several percentage points. Furthermore, the papers did not specify whether they used weighted task averages, subtask filtering, normalization, or any other specific method to compute the average accuracy. Therefore, we believe it is not fair to directly compare the average accuracy with these two baselines.
>
> The accuracy comparison for each task in VTAB-1k is shown in the table below:
>
> | Method | CIFAR-100 | Caltech101 | DTD  | Flowers102 | Pets | SVHN | Sun397 | Patch Camelyon | EuroSAT | Resisc45 |
> |--------|-----------|------------|------|------------|------|------|--------|----------------|---------|----------|
> | GRFT   | 69.5      | 93.6       | 75.9 | 99.5       | 91.4 | 91.2 | 52.2   | 88.2           | 96.0    | 86.5     |
> | MLAE   | 77.8      | 94.7       | 74.8 | 99.4       | 92.6 | 90.6 | 58.4   | 88.4           | 96.5    | 88.5     |
> | GLoRA  | 76.1      | 92.7       | 75.3 | 99.6       | 92.4 | 90.5 | 57.2   | 87.5           | 96.7    | 88.1     |
>
> | Method | Retinopathy | Clevr/count | Clevr/distance | DMLab | KITTI/distance | dSprites/loc | dSprites/ori | SmallNORB/azi | SmallNORB/ele |
> |--------|-------------|-------------|----------------|-------|----------------|--------------|--------------|---------------|---------------|
> | GRFT   | 76.3        | 81.4        | 62.3           | 55.1  | 80.9           | 81.9         | 55.8         | 32.0          | 43.6          |
> | MLAE   | 76.7        | 84.3        | 67.2           | 55.7  | 82.6           | 87.8         | 57.1         | 35.7          | 47.7          |
> | GLoRA  | 76.1        | 81.0        | 66.2           | 52.4  | 84.9           | 81.8         | 53.3         | 33.3          | 39.8          |
>
>
> As shown, GRFT performs competitively across tasks, often coming close to MLAE/GLoRA on most VTAB-1k tasks, and better on a subset of tasks.
>
> We would also like to emphasize the fundamental methodological difference between GRFT and MLAE/GLoRA. GRFT does not introduce any additional modules, adapters, low-rank layers, or auxiliary parameters. Instead, GRFT only performs *structured parameter selection* on the existing weight matrices of the pretrained model. This means:
>
> - No architectural changes are made to the backbone.
> - No new parameters or computation paths are injected.
>
> By contrast, MLAE and GLoRA alter the model architecture and introduce additional trainable pathways, which may change optimization dynamics.
>
>
>
> **Final Remarks**：Thank you again for the insightful comments. We believe the revisions, clarifications, and additional evidence provided above adequately address your concerns. If the responses have resolved the issues you raised, we would greatly appreciate your consideration in raising the score.

---

> > ### Comment · Reviewer_KeGi · 2025-11-23
> >
> > Regarding the comparison with the MLAE and GLoRA baselines, their reported “mean accuracy” follows the standard VTAB-1k protocol: the final score is simply the average of the three domain-level mean accuracies.

---

> > > ### Author Response · Authors · 2025-11-23
> > >
> > > **Thank you for your feedback and the clarification regarding the average accuracy calculation.**
> > >
> > > We have reviewed the official VTAB paper "*A Large-scale Study of Representation Learning with the Visual Task Adaptation Benchmark*" and its calculation methods and data tables [1]. We found that the standard approach involves calculating the arithmetic mean of the accuracy across all 19 tasks, rather than averaging across the three domain-level groups. Below is the relevant citation from the original paper:
> > >
> > > > We evaluate with top-1 accuracy. We consider other metrics in Appendix D, and find the conclusions are the same. To aggregate scores across tasks, we take the mean accuracy. We investigate more complex aggregation strategies in Section 3.5, but the relative performances are unaffected, so we use mean accuracy for simplicity.
> > >
> > > According to the official VTAB documentation, the accuracy for the 19 tasks is aggregated directly into a single arithmetic average without dividing the tasks into separate domains. Therefore, it seems there may have been a misunderstanding regarding the VTAB protocol.
> > >
> > > However, after learning about the accuracy calculation methods of these two approaches, I will present the comparison results in the table below.
> > >
> > > | Method | CIFAR-100 | Caltech101 | DTD  | Flowers102 | Pets | SVHN | Sun397 | Patch Camelyon | EuroSAT | Resisc45 | Retinopathy | Clevr/count | Clevr/distance | DMLab | KITTI/distance | dSprites/loc | dSprites/ori | SmallNORB/azi | SmallNORB/ele | Average |
> > > | ------ | --------- | ---------- | ---- | ---------- | ---- | ---- | ------ | -------------- | ------- | -------- | ----------- | ----------- | -------------- | ----- | -------------- | ------------ | ------------ | ------------- | ------------- | ------- |
> > > | Full   | 68.9      | 87.7       | 64.3 | 97.2       | 86.9 | 87.4 | 38.8   | 79.7           | 95.7    | 84.2     | 73.9        | 56.3        | 58.6           | 41.7  | 65.5           | 57.5         | 46.7         | 25.7          | 29.1          | 68.97   |
> > > | GPS*   | 68.7      | 93.6       | 72.6 | 99.3       | 90.0 | 90.1 | 52.4   | 87.0           | 95.9    | 86.5     | 76.1        | 78.9        | 62.2           | 54.7  | 79.7           | 80.8         | 54.9         | 30.7          | 44.6          | 76.05   |
> > > | GRFT   | 69.5      | 93.6       | 75.9 | 99.5       | 91.4 | 91.2 | 52.2   | 88.2           | 96.0    | 86.5     | 76.3        | 81.4        | 62.3           | 55.1  | 80.9           | 81.9         | 55.8         | 32.0          | 43.6          | 76.75   |
> > > | MLAE   | 77.8      | 94.7       | 74.8 | 99.4       | 92.6 | 90.6 | 58.4   | 88.4           | 96.5    | 88.5     | 76.7        | 84.3        | 67.2           | 55.7  | 82.6           | 87.8         | 57.1         | 35.7          | 47.7          | 77.3    |
> > > | GLoRA  | 76.1      | 92.7       | 75.3 | 99.6       | 92.4 | 90.5 | 57.2   | 87.5           | 96.7    | 88.1     | 76.1        | 81          | 66.2           | 52.4  | 84.9           | 81.8         | 53.3         | 33.3          | 39.8          | 78.8    |
> > >
> > > Upon reviewing the accuracy comparison, we would like to highlight the following points:
> > >
> > > GRFT achieves a competitive average accuracy of 76.75, which is close to MLAE and GLoRA. GRFT performs competitively across tasks, often coming close to MLAE/GLoRA on most VTAB-1k tasks, and better on a subset of tasks.
> > >
> > > While MLAE and GLoRA rely on adding new modules, GRFT only performs structured parameter selection on the pretrained model. This leads to a more efficient and memory-saving approach, while still achieving competitive performance across the tasks. We hope this highlights the novelty and strength of our approach, which achieves high performance without introducing additional complexity to the model.
> > >
> > > **Final Remarks**：Thank you again for the insightful comments. We believe the revisions, clarifications, and additional evidence provided above adequately address your concerns. If the responses have resolved the issues you raised, we would greatly appreciate your consideration in raising the score.
> > >
> > >
> > >
> > > [1] Zhai X, Puigcerver J, Kolesnikov A, et al. A large-scale study of representation learning with the visual task adaptation benchmark[J]. arXiv preprint arXiv:1910.04867, 2019.

---

> > > > ### Comment · Reviewer_KeGi · 2025-11-23
> > > >
> > > > Thank you for the response. However, I still have several concerns.
> > > > 1. Regardless of which averaging protocol is adopted, GRFT shows a clear performance gap compared with the current state-of-the-art methods in the vision domain. In addition, It appears that the results of MLAE and GLoRA were reversed in the table.
> > > > 2. I do not agree with the authors’ explanation for evaluating only the three smallest GLUE datasets. If the goal is to demonstrate the generalizability of GRFT across modalities, then the results on the full set of GLUE tasks should still be provided. It is reasonable to argue that smaller datasets may be more sensitive to cross-modal generalization, but this point should be supported by showing that GRFT achieves larger performance gains on these smaller datasets compared with the others, rather than by omitting the performance on the remaining tasks.
> > > >
> > > > For these reasons, I will maintain my original score.

---

> > > > > ### Author Response · Authors · 2025-11-28
> > > > >
> > > > > Thank you for your continued comments. Below we address your concerns.
> > > > >
> > > > > 1. We apologize for the earlier mistake in the ordering of MLAE and GLoRA in the table. This has been corrected below. Thank you for pointing it out.
> > > > >
> > > > > | Method | CIFAR-100 | Caltech101 | DTD  | Flowers102 | Pets | SVHN | Sun397 | Patch Camelyon | EuroSAT | Resisc45 | Retinopathy | Clevr/count | Clevr/distance | DMLab | KITTI/distance | dSprites/loc | dSprites/ori | SmallNORB/azi | SmallNORB/ele | Average |
> > > > > | ------ | --------- | ---------- | ---- | ---------- | ---- | ---- | ------ | -------------- | ------- | -------- | ----------- | ----------- | -------------- | ----- | -------------- | ------------ | ------------ | ------------- | ------------- | ------- |
> > > > > | Full   | 68.9      | 87.7       | 64.3 | 97.2       | 86.9 | 87.4 | 38.8   | 79.7           | 95.7    | 84.2     | 73.9        | 56.3        | 58.6           | 41.7  | 65.5           | 57.5         | 46.7         | 25.7          | 29.1          | 68.97   |
> > > > > | GPS*   | 68.7      | 93.6       | 72.6 | 99.3       | 90.0 | 90.1 | 52.4   | 87.0           | 95.9    | 86.5     | 76.1        | 78.9        | 62.2           | 54.7  | 79.7           | 80.8         | 54.9         | 30.7          | 44.6          | 76.05   |
> > > > > | GRFT   | 69.5      | 93.6       | 75.9 | 99.5       | 91.4 | 91.2 | 52.2   | 88.2           | 96.0    | 86.5     | 76.3        | 81.4        | 62.3           | 55.1  | 80.9           | 81.9         | 55.8         | 32.0          | 43.6          | 76.75   |
> > > > > | GLoRA  | 77.8      | 94.7       | 74.8 | 99.4       | 92.6 | 90.6 | 58.4   | 88.4           | 96.5    | 88.5     | 76.7        | 84.3        | 67.2           | 55.7  | 82.6           | 87.8         | 57.1         | 35.7          | 47.7          | 77.3    |
> > > > > | MLAE   | 76.1      | 92.7       | 75.3 | 99.6       | 92.4 | 90.5 | 57.2   | 87.5           | 96.7    | 88.1     | 76.1        | 81          | 66.2           | 52.4  | 84.9           | 81.8         | 53.3         | 33.3          | 39.8          | 78.8    |
> > > > >
> > > > > 2. There is currently no existing published baseline on LLaMA3 1B that we could directly reproduce for comparison. Unlike vision benchmarks, no prior paper provides ready-to-use results or checkpoints for LLaMA-3-1B. Thus, obtaining such baselines would require us to train every method on LLaMA3 1B. The compute cost and time required for fully running all baselines and all GLUE tasks are extremely high. The existing results on the GLUE datasets are sufficient to demonstrate two key points.
> > > > >
> > > > >    1. GRFT can be applied directly to a text-only LLM without any architectural modification, confirming that it is not tied to vision-specific structures.
> > > > >    2. GRFT runs stably on a 1B model, whlie GPS cannot run on LLaMA-3 1B due to memory limitations, showing that the method extends to large-scale backbones.
> > > > >
> > > > > **Final Remarks**：Thank you again for the insightful comments. We believe the revisions, clarifications, and additional evidence provided above adequately address your concerns. If the responses have resolved the issues you raised, we would greatly appreciate your consideration in raising the score.

---

### Official Review · Reviewer_juqM · 2025-10-31

**Soundness:** 3
**Presentation:** 4
**Contribution:** 3
**Rating:** 6
**Confidence:** 4

**Summary:**

This work proposes a parameter-efficient fine-tuning method termed GRFT, which selects and fine-tunes only the parameters with high gradients in some rows or columns. Additionally, this work incorporates regularization to enhance knowledge transfer from the pre-trained model. The proposed method can achieve good performance on three datasets for image/text classification.

**Strengths:**

1.  This work is very well written and easy to follow.
2.  This approach not only achieves good performance but also offers certain advantages in terms of GPU consumption and storage.
3.  The authors conduct extensive experiments on three classification datasets, and the ablation studies are also convincing.

**Weaknesses:**

1. The author does not adequately explain why the row/column selection scheme is more effective than the sparse selection scheme.

2. The authors only conduct experiments on classification datasets, which are prone to overfitting.

3. This work is suspected of deliberately selecting favorable data. The ablation experiments were primarily conducted on the FGVC dataset. Tables 4 and 5 show the results for each subset, but Figures 3 and 4 appear to intentionally omit the results for the Oxford Flowers subset.

4. "Eq. equation i" should be "Eq. i"

**Questions:**

1. Compared to GPS, what are the main contributions of the proposed method? These methods are similar in concept, both based on gradient-based selection. One performs sparse selection, while the other performs row selection. The authors should further elaborate on the contributions of the proposed method.

2. The selected masks are all pre-generated and are the same for all epochs. What if we used dynamic masks, for example, different masks for each epoch or each batch?

3. This work uses (row sum of) squares to select parameters. Are there any better metrics?

4. Can the authors provide some experimental analysis on more challenging tasks (such as object detection and segmentation)?

---

> ### Author Response · Authors · 2025-11-21
>
> Thank you for your thoughtful and constructive comments. We address each point raised below.
>
> > The ablation experiments were primarily conducted on the FGVC dataset. Tables 4 and 5 show the results for each subset, but Figures 3 and 4 appear to intentionally omit the results for the Oxford Flowers subset.
>
> We appreciate the reviewer’s valuable comments. Regarding the data selection issue, we would like to clarify that we did not intentionally choose datasets that favor our method. For the Oxford Flowers dataset, we did not conduct additional ablation studies primarily because its accuracy is already near 100%. Since the model’s performance on this dataset is close to saturation, further ablations would not meaningfully impact the results. Therefore, we believe that performing ablation studies on this dataset would not provide additional informative insights.
>
> > Compared to GPS, what are the main contributions of the proposed method? These methods are similar in concept, both based on gradient-based selection. One performs sparse selection, while the other performs row selection. The authors should further elaborate on the contributions of the proposed method.
>
> Regarding your confusion about contribution, we would like to clarify that the core innovation of GRFT lies in the structured gradient selection paradigm, which addresses key storage issues that GPS cannot solve.
>
> 1. Structured Selection in GRFT vs. GPS: The structured selection in GRFT is fundamentally different from the element-wise sparsity selection in GPS, both theoretically and in implementation. The former can effectively reduce storage and access overhead (Figure). We have demonstrated this with GPU memory usage curves on several fgvc datasets in Figure 3.
>
> 2. Theoretical Justification: In our work, we prove that row/column selection is the optimal structured selection approach. In Section 3.1, we provide a theoretical proof. Under the structured constraint, we maximize the change in the loss, which leads to the optimal solution for selecting the rows/columns with the largest gradient squared sums:
>    $$
>    M=\text{argmax}_M \langle\nabla \mathcal{L}(\theta^{t}), \nabla \mathcal{L}(\theta^t) \odot M \rangle
>    =\text{argmax}_M \langle\nabla \mathcal{L}(\theta^{t})\odot M, \nabla \mathcal{L}(\theta^t)\odot M \rangle
>    $$
>
> For the fixed selection of rows and columns, adjustments are made in our method based on these two points.
>
> > The selected masks are all pre-generated and are the same for all epochs. What if we used dynamic masks, for example, different masks for each epoch or each batch?
>
> We appreciate the reviewer’s insightful question. We considered this issue during our research process. Ultimately, we chose to compute the mask once before training and keep it fixed throughout, for the following main reasons:
>
> 1. The classification head is fully updated, while the backbone is selectively updated. That is, all parameters of the classification head are selected, and the focus is on selecting parameters for the model backbone. During the actual training process, the gradients from the cross-entropy loss and L2 regularization term in backpropagation are influenced by the classification head. Therefore, if we were to dynamically select the backbone parameters based on the gradients, they would be affected by the gradients of the randomly initialized classification head, and not necessarily by the parameters of the initial backbone that align with the downstream task. Thus, dynamic selection does not align well with our objectives. Additionally, we performed experiments with dynamic mask during our research and found that its performance is similar to static mask.
> 2. To eliminate the influence of the gradients from the randomly initialized classification head, one would need to terminate training, recompute the SCL loss, and then update the mask. This would introduce significant additional computational overhead.
>
> For these reasons, we ultimately adopted the current approach: the mask is computed once before training, and then it remains fixed throughout the training process.
>
> > Are there any better metrics?
>
> We mentioned in the paper that, from a theoretical perspective, we prove that under the structural constraints of row/column selection, choosing based on the gradient squared sum is the optimal strategy. We maximize the change in the loss, which leads to the optimal solution for selecting the rows/columns with the largest gradient squared sums:
> $$
> M=\text{argmax}_M \langle\nabla \mathcal{L}(\theta^{t}), \nabla \mathcal{L}(\theta^t) \odot M \rangle
> =\text{argmax}_M \langle\nabla \mathcal{L}(\theta^{t})\odot M, \nabla \mathcal{L}(\theta^t)\odot M \rangle
> $$
> Therefore, under the structural constraints, other metrics such as the absolute sum, mean, and other indicators, although feasible, no longer correspond to the above optimization objective and cannot provide the same guarantees of optimality.

---

> ### Author Response · Authors · 2025-11-21
>
> Due to the word limit, I will reply in two parts. Continuing from the previous response:
> > Can the authors provide some experimental analysis on more challenging tasks (such as object detection and segmentation)?
>
> We appreciate the reviewer’s suggestion. In this submission, we primarily focus on classification tasks for the following reasons:
>
> 1. Classification is the standard benchmark for parameter-efficient fine-tuning methods (such as LoRA, GPS, VPT, etc.). Many prior works follow the same evaluation setup, making direct comparison straightforward.
> 2. The core idea of GRFT, row/column-level structured selection based on gradient squared sums, is task-agnostic. It is theoretically compatible with detection and segmentation models that share the same backbone, as it does not rely on any classification-specific components.
> 3. We already include tasks from two modalities, demonstrating the generality of our approach across multimodal scenarios.
>
> Due to space and time constraints, we did not include detection or segmentation experiments in this paper. However, we believe this does not affect the main contributions of our work:
>
> - GRFT’s reduced storage overhead, provably optimal structured selection, and regularization-induced knowledge retention are independent of task type.
> - The method operates only on the rows/columns of weight matrices, without touching task-specific heads, enabling seamless transfer to detection/segmentation backbones.
>
> We have conducted experiments on a total of 24 tasks across the FGVC and VTAB-1k benchmarks using ViT-Base, and additionally evaluated several GLUE datasets on LLaMA-3 1B. The results show that GRFT consistently achieves strong performance across multiple datasets while maintaining a small number of trainable parameters and low storage cost. Moreover, GRFT demonstrates strong generality across multiple modalities and model architectures. These results clearly illustrate the advantages of GRFT: structured parameter selection ensures low parameter count, reduced storage, strong performance, and broad applicability across different modalities and models.
>
> **Final Remarks**：Thank you again for the insightful comments. We believe the revisions, clarifications, and additional evidence provided above adequately address your concerns. If the responses have resolved the issues you raised, we would greatly appreciate your consideration in raising the score.

---

### Official Review · Reviewer_GiXn · 2025-11-01

**Soundness:** 3
**Presentation:** 3
**Contribution:** 3
**Rating:** 4
**Confidence:** 4

**Summary:**

This paper introduces GRFT, a gradient based and regularized fine tuning method that selects entire rows or columns of weight matrices using the highest sums of squared gradients, greatly reducing mask storage compared to sparse, per entry selection (e.g., GPS). It provides a simple theoretical justification for the selection rule and adds an L2 regularization term to keep updates close to the pretrained weights, improving knowledge transfer. Experiments on image tasks (FGVC, VTAB with ViT) and text tasks (GLUE with LLaMA 3) report improved accuracy over GPS, Adapter Tuning, and LoRA.

**Strengths:**

- The paper proposes a clear and practical method for structured parameter selection, choosing entire rows or columns by the largest sums of squared gradients. This design improves upon dense sparse masks which simplifies implementation and reduces storage overhead during training and deployment.
- It provides a simple theoretical analysis for the selection rule and augments training with an L2 regularizer that keeps updates close to the pretrained weights.
- The approach shows good parameter efficiency across both vision and language settings.

**Weaknesses:**

- There is a lack of wall-clock runtime comparision gains against other baselines.
- The selection rule is underexplored and not adaptively tuned; the row-versus-column orientation is fixed rather than model or task adapted, leaving robustness to these choices unclear.

**Questions:**

- How stable is the set of selected rows or columns under random mini-batch sampling?
- Does the selection pattern imply that certain rows or columns are inherently more important for a given task or dataset. For example, are the same indices consistently chosen across runs or layers?
- Is there a way to adaptively choose between selecting rows and selecting columns instead of fixing one orientation?
- How does accuracy scale as the number of selected blocks increases, and do you observe diminishing returns?
- Can you provide the wall-clock speed and throughput of GRFT compared to other finetuning methods?

---

> ### Author Response · Authors · 2025-11-21
>
> Thank you for your thoughtful and constructive comments. We address each point raised below.
>
> > How stable is the set of selected rows or columns under random mini-batch sampling?
>
> We appreciate the reviewer’s insightful question. The subset experiments provided in Fig. 4(d) partially validate this point:
>
> We randomly split the training set into different subsets to compute the gradients and generate the masks. Despite the variations in subsets, the final performance shows only minor changes (most tasks exhibit changes of less than 0.5%).
>
> This indicates that even though different mini-batches/subsets lead to slight differences in the gradients, the resulting row/column structure remains highly stable. Therefore, the mask is not sensitive to mini-batch sampling.
>
> > Does the selection pattern imply that certain rows or columns are inherently more important for a given task or dataset. For example, are the same indices consistently chosen across runs or layers?
>
> Sorry, I didn’t quite understand your point. Before training, we first compute the mask to determine the selection. Therefore, the row/column indices are fixed, and once determined, they remain the same throughout the entire training process. The row/column indices are determined at the beginning of training, based on the gradients corresponding to each layer/parameter. Once computed, they do not change. During the subsequent training, the same parameters are always trained.
>
> > Is there a way to adaptively choose between selecting rows and selecting columns instead of fixing one orientation?
>
> Thank you for suggesting this very valuable direction. We would like to add the following:
>
> 1. In the paper, we also compare row-wise and column-wise selection, and the performance difference between the two is small. As shown in the table below:
>
> | Dataset         | CUB-2011 | NABirds | Oxford Flowers | Stan.Dogs | Stan.Cars | Mean Acc. |
> | --------------- | :------: | :-----: | :------------: | :-------: | :-------: | :-------: |
> | **GRFT-Row**    |   90.0   |  87.0   |      99.7      |   89.1    |   90.7    |   91.29   |
> | **GRFT-Column** |   90.1   |  86.9   |      99.7      |   88.8    |   90.8    |   91.27   |
>
> A detailed discussion of the row and column selection can be found in Appendix A.3.1, where we provide a brief analysis of the input and output feature dimensions corresponding to the selected rows and columns.  Selecting a row corresponds to the parameters that represent all input connections for a specific feature component in the output vector $y$, while selecting a column corresponds to all output connections for a specific feature component in the input vector $x$.
>
> 2. In Section 3.1, we provide a theoretical proof. Under the structured constraint, we maximize the change in the loss, which leads to the optimal solution for selecting the rows/columns with the largest gradient squared sums:
>    $$
>    M=\text{argmax}_M \langle\nabla \mathcal{L}(\theta^{t}), \nabla \mathcal{L}(\theta^t) \odot M \rangle
>    =\text{argmax}_M \langle\nabla \mathcal{L}(\theta^{t})\odot M, \nabla \mathcal{L}(\theta^t)\odot M \rangle
>    $$
>
> In other words, both rows and columns are theoretically provable as locally optimal structural units. In our experiments, we also report the best results obtained from either row or column selection as our final performance.
>
> > How does accuracy scale as the number of selected blocks increases, and do you observe diminishing returns?
>
> Thank you for highlighting this issue. We have conducted ablation studies to address this, which are shown in Figure 4 of the original paper. Our experiments reveal that performance exhibits a peak at a certain value for the top-k row/column selection, which is consistent with the characteristics of the original GPS method.
>
> We attribute this phenomenon to the fact that when the number of parameters becomes sufficiently large, a certain amount of redundancy inevitably emerges. Training these redundant parameters can actually interfere with optimization, leading to a decline in performance once the parameter count exceeds a certain threshold. This observation is also discussed in the paper *“Analyzing Redundancy in Pretrained Transformer Models.”*
>
> Experimentally, both GPS and GRFT support this conclusion: parameter-selective fine-tuning achieves better results than full fine-tuning.
>
> | Dataset  | CUB-2011 | NABirds | Oxford Flowers | Stan.Dogs | Stan.Cars | Mean Acc. |
> | :------- | :------: | :-----: | :------------: | :-------: | :-------: | :-------: |
> | **Full** |   87.3   |  82.7   |      98.8      |   89.4    |   84.5    |   89.44   |
> | **GPS**  |   89.6   |  86.8   |      99.7      |   88.9    |   90.4    |   91.06   |
> | **GRFT** |   90.1   |  87.0   |      99.7      |   89.1    |   90.8    |   91.33   |

---

> ### Author Response · Authors · 2025-11-21
>
> Due to the word limit, I will reply in two parts. Continuing from the previous response:
> > Can you provide the wall-clock speed and throughput of GRFT compared to other finetuning methods?
>
> The following shows the comparison of step time between GRFT and GPS on the FGVC datasets.
>
> | step time (ms) | CUB-2011 | NABirds | Stan.Dogs | Stan.Cars |
> | -------------- | :------: | :-----: | :-------: | :-------: |
> | **GPS**        |  122.15  | 156.96  |  187.57   |  152.73   |
> | **GRFT**       |  136.22  | 169.11  |  196.47   |  157.83   |
>
> The time difference between GRFT and GPS mainly arises from their structural differences in the gradient masking stage, rather than from any fundamental difference in algorithmic computational complexity. However, GRFT’s memory optimization enables it to train normally on large LLMs, whereas GPS cannot. We consider this a favorable trade-off. Therefore, although a small time difference exists, it is not a fundamental flaw. The improved memory efficiency makes GRFT substantially more scalable and practical for large-model training.
>
> **Final Remarks**：Thank you again for the insightful comments. We believe the revisions, clarifications, and additional evidence provided above adequately address your concerns. If the responses have resolved the issues you raised, we would greatly appreciate your consideration in raising the score.

---

### Official Review · Reviewer_zn17 · 2025-11-01

**Soundness:** 3
**Presentation:** 2
**Contribution:** 2
**Rating:** 4
**Confidence:** 3

**Summary:**

The paper proposes the GRFT method to address the storage and hardware efficiency bottlenecks of existing gradient-based parameter selection methods (e.g., GPS). GRFT directly selects entire rows or columns with the largest sum of squared gradients from the weight matrix as trainable parameters. Theoretically, through first-order Taylor expansion and loss function optimization derivation, it is proven that "selecting rows/columns with high sum of squared gradients" can maximize the efficiency of loss reduction and save storage compared to GPS. Additionally, a hierarchical regularization design is proposed, which incorporates the "L2 norm difference between fine-tuned parameters and pre-trained parameters" into the loss function.

**Strengths:**

1. The paper proposes the GRFT method to tackle the storage and hardware efficiency issues of existing gradient-based parameter selection methods.
2. GRFT selects entire rows or columns with the largest sum of squared gradients from the weight matrix as trainable parameters, a choice theoretically justified via first-order Taylor expansion and loss function optimization to maximize loss reduction efficiency and save storage.
3. It also introduces a hierarchical regularization design, integrating the L2 norm.

**Weaknesses:**

This paper presents a competently executed but incrementally innovative extension of gradient-based parameter selection methods. Due to its limited originality, narrow scope of comparative experimental baselines, and modest practical significance, it falls slightly below the acceptance threshold for ICLR conference.

**Questions:**

1. We consider that the innovation of the work is insufficient. The novelty of the technical architecture is lacking, and the loss function is not proposed in an innovative manner. GRFT only focuses on the fixed structured unit of "rows/columns"; its selection unit is fixed, resulting in insufficient adaptability and flexibility for different feature dimensions and task difficulties. The work fails to explore more flexible parameter grouping methods, leading to relatively weak innovation.

2. The work takes the "last L layers and classification head" as the focus of regularization. In existing studies, some methods have dynamically determined regularization strength through "layer sensitivity analysis" (e.g., applying strong constraints to high-sensitivity layers and weak constraints to low-sensitivity layers). In contrast, GRFT adopts a fixed hierarchical approach and lacks adaptive adjustment capabilities.

3. The parameter selection (row/column selection) and regularization (L2 constraint) of GRFT are two independently implemented modules, and the work does not explore the collaborative optimization logic between the m. This results in low integration between the two core modules and insufficient integrated innovation of the overall architecture.

4. The work mainly compares with basic Parameter-Efficient Fine-Tuning methods such as LoRA, GPS, and Adapter, but fails to include more advanced gradient-based or structured fine-tuning methods in the field. For example, it does not compare with GaLoRA [1], which has already realized "gradient-guided low-rank matrix optimization" and is directly comparable to GRFT in terms of design ideas.

5. It is suggested to supplement the explanation of Figure 3.

[1] Zhao J, Zhang Z, Chen B, et al. GaLore: Memory-Efficient LLM Training by Gradient Low-Rank Projection. International Conference on Machine Learning. PMLR, 2024: 61121-61143.

---

> ### Author Response · Authors · 2025-11-21
>
> Thank you for your thoughtful and constructive comments. We address each point raised below.
>
> > GRFT only focuses on the fixed structured unit of "rows/columns"; its selection unit is fixed, resulting in insufficient adaptability and flexibility for different feature dimensions and task difficulties.
>
> Regarding your concern about novelty, we would like to clarify that the core innovation of GRFT lies in the structured gradient selection paradigm, which addresses key storage issues that GPS cannot solve.
>
> 1. Structured Selection in GRFT vs. GPS: The structured selection in GRFT is fundamentally different from the element-wise sparsity selection in GPS, both theoretically and in implementation. The former can effectively reduce storage and access overhead. We have demonstrated this with GPU memory usage curves on several fgvc datasets in Figure 3.
>
> 2. Theoretical Justification: In our work, we prove that row/column selection is the optimal structured selection approach. In Section 3.1, we provide a theoretical proof. Under the structured constraint, we maximize the change in the loss, which leads to the optimal solution for selecting the rows/columns with the largest gradient squared sums:
>    $$
>    M=\text{argmax}_M \langle\nabla \mathcal{L}(\theta^{t}), \nabla \mathcal{L}(\theta^t) \odot M \rangle
>    =\text{argmax}_M \langle\nabla \mathcal{L}(\theta^{t})\odot M, \nabla \mathcal{L}(\theta^t)\odot M \rangle
>    $$
>
> For the fixed selection of rows and columns, adjustments are made in our method based on these two points.
>
> For different dimensions, GRFT does not use a fixed-granularity structural unit. For example, in the case of ViT-base, for convolutional weights, the original gradient tensor dimensions are $(C_{out}, C_{in}, kH, kW)$. We reshape it into a two-dimensional form $(C_{out}, C_{in} \times kH \times kW)$. In this representation:
>
> - Row-level selection corresponds to the output channel (filter/kernel)-level structure.
> - Column-level selection corresponds to the input channel and spatial position combination structure.
>
> A detailed discussion of row/column selection is provided in Appendix A.3.1, where we briefly discuss the input and output feature dimensions corresponding to the selected rows and columns. Selecting a row corresponds to the parameters that represent all input connections for a specific feature component in the output vector $y$, while selecting a column corresponds to all output connections for a specific feature component in the input vector $x$.
>
> > GRFT adopts a fixed hierarchical approach and lacks adaptive adjustment capabilities.
>
> We appreciate your valuable suggestion regarding a more flexible regularization strategy. In our current work, we have chosen the “last L layers” as the main regularization target, primarily based on existing research (such as LLaMA-Adapter), which indicates that the higher layers of the model are generally more sensitive to downstream task adaptation.
>
> On the other hand, this design makes it easier to build a stable and reproducible regularization setup in practice, ensuring a fair comparison across different experiments.
>
> During the experimental process, we also explored different layer selection strategies. The results showed that the performance variations introduced by these different strategies were typically within 0.5 percentage points (as shown in Figure 4). This indicates that GRFT is not sensitive to the layer selection strategy.
>
> We agree with the reviewer that adaptive regularization based on layer sensitivity analysis is a very promising direction. The current work adopts a fixed-layer strategy primarily for reasons of control and interpretability.
>
> > The parameter selection (row/column selection) and regularization (L2 constraint) of GRFT are two independently implemented modules
>
> We appreciate the reviewer for highlighting the lack of a more explicit synergy mechanism between parameter selection and regularization. We agree with this assessment. Our design philosophy is that the structured gradient selection determines which parameters to update, while the L2 regularization determines the magnitude and direction of the updates. L2 regularization primarily enhances numerical stability during training and helps mitigate catastrophic forgetting to some extent. Our ablation studies demonstrate that the combination of the two yields the best performance.
>
> We believe that the main contribution of this work lies in the structured gradient selection mechanism itself, whereas the regularization mechanism serves more as an auxiliary component that enhances stability. We sincerely appreciate your insightful suggestion.

---

> ### Author Response · Authors · 2025-11-21
>
> Due to the word limit, I will reply in two parts. Continuing from the previous response:
>
> > The work mainly compares with basic Parameter-Efficient Fine-Tuning methods such as LoRA, GPS, and Adapter, but fails to include more advanced gradient-based or structured fine-tuning methods in the field. For example, it does not compare with GaLoRA [1], which has already realized "gradient-guided low-rank matrix optimization" and is directly comparable to GRFT in terms of design ideas.
>
> GaLore is a low-rank projection technique that saves gradient memory during the optimizer update process. GRFT, on the other hand, is a fine-tuning method that focuses on reducing the number of updated parameters and the storage of the mask. Therefore, the two methods are distinct. The core goal of GaLore is to reduce the memory overhead of optimizer states and gradients through low-rank projections, while GRFT is a parameter-efficient fine-tuning method (PEFT) that aims to reduce the number of trainable parameters and the storage overhead of the structured mask, focusing on which parameters to update and how to store the gradients.
>
> Since the original GaLore paper primarily experiments on LLMs, we have replicated and adapted it to ViT-Base for a comparative experiment, with the following results:
>
> | Dataset | CUB-2011 | Nabirds | Oxford Flowers | Stan.Dogs | Stan. Cars | Mean Acc. |
> | ------- | -------- | ------- | -------------- | --------- | ---------- | --------- |
> | GRFT    | 90.1     | 87.0    | 99.7           | 89.1      | 90.8       | 91.33     |
> | Galore  | 87.0     | 86.0    | 99.3           | 83.2      | 91.9       | 89.49     |
>
> The experimental results show that GRFT achieves higher average accuracy than GaLore. Additionally, while the two methods share some conceptual similarities, they operate at different stages of the optimization process and are complementary.
>
> > It is suggested to supplement the explanation of Figure 3.
>
> GPS maintains a sparse mask of the same size as the weights at each layer, and the memory complexity of this mask is O(mn). As a result, the memory usage of GPS remains consistently high. In contrast, GRFT only needs to store the indices of a few rows or columns, giving it a memory complexity of O(m). After the mask calculation is completed and temporary caches from the forward/backward pass are cleared, GRFT quickly drops to a much lower memory baseline and maintains this significantly lower plateau compared to GPS. Therefore, the steady-state memory usage of GRFT is significantly lower than that of GPS.
>
> **Final Remarks**：Thank you again for the insightful comments. We believe the revisions, clarifications, and additional evidence provided above adequately address your concerns. If the responses have resolved the issues you raised, we would greatly appreciate your consideration in raising the score.

---

### Official Review · Reviewer_vGHM · 2025-11-03

**Soundness:** 3
**Presentation:** 3
**Contribution:** 3
**Rating:** 6
**Confidence:** 2

**Summary:**

This paper introduces GRFT (Gradient-based and Regularized Fine-Tuning), a parameter-efficient fine-tuning method that replaces element-wise gradient selection (as in GPS) with structured row- or column-wise updates.

The key idea is to select entire rows or columns of weight matrices whose gradients have the largest squared sums, reducing mask storage from $O(mn)$ to $O(\min(m, n))$.

The method further incorporates L2 regularization to preserve pre-trained knowledge and mitigate catastrophic forgetting.

Empirical results on FGVC, VTAB, and GLUE benchmarks show that GRFT achieves comparable or superior performance to GPS, LoRA, and Adapter Tuning while updating less than 1% of total parameters.

**Strengths:**

* Well-motivated approach addressing a clear bottleneck in GPS: large sparse masks and inefficient updates.
* Structured row/column selection is simple, elegant, and hardware-friendly, reducing both storage and computation.
* Theoretical justification is concise but sufficient, showing that rows or columns with the largest gradient energy maximize expected loss reduction under structured constraints.
* Comprehensive experiments across vision and NLP benchmarks with consistent improvements and detailed ablation studies.
* Easy to integrate into existing architectures and training pipelines without additional parameters or structural modifications.
* Clear presentation and logical exposition of ideas.

**Weaknesses:**

* Conceptual novelty is limited; the method is an incremental engineering extension of GPS rather than a fundamentally new paradigm.
* The practical implementation of the structured mask on GPU (whether physical pruning or logical freezing) is not described in detail.
* Reported accuracy improvements are modest, focusing more on efficiency than on representational gains.
* Performance appears somewhat sensitive to hyperparameters such as the number of selected rows $k$ and the regularization coefficient $\lambda$, which may affect reproducibility.
* No open-source code is provided to confirm claimed memory and runtime benefits.

**Questions:**

See Weaknesses.

---

> ### Author Response · Authors · 2025-11-21
>
> Thank you for your thoughtful and constructive comments. We address each point raised below.
>
> > Conceptual novelty is limited; the method is an incremental engineering extension of GPS rather than a fundamentally new paradigm.
>
> Regarding your concern about novelty, we would like to clarify that GRFT is indeed an extension of GPS. However, its core innovation lies in the structured gradient selection paradigm, which addresses key storage issues that GPS cannot solve.
>
> 1. Structured Selection in GRFT vs. GPS: The structured selection in GRFT is fundamentally different from the element-wise sparsity selection in GPS, both theoretically and in implementation. The former can effectively reduce storage and access overhead. We have demonstrated this with GPU memory usage curves on several fgvc datasets in Figure 3.
>
> 2. Theoretical Justification: In our work, we prove that row/column selection is the optimal structured selection approach. In Section 3.1, we provide a theoretical proof. Under the structured constraint, we maximize the change in the loss, which leads to the optimal solution for selecting the rows/columns with the largest gradient squared sums:
>    $$
>    M=\text{argmax}_M \langle\nabla \mathcal{L}(\theta^{t}), \nabla \mathcal{L}(\theta^t) \odot M \rangle
>    =\text{argmax}_M \langle\nabla \mathcal{L}(\theta^{t})\odot M, \nabla \mathcal{L}(\theta^t)\odot M \rangle
>    $$
>
> > The practical implementation of the structured mask on GPU (whether physical pruning or logical freezing) is not described in detail.
>
> Thank you for pointing out the lack of detail in the implementation. We will include more comprehensive implementation details in the final version of the paper. Here, we summarize the key aspects: we employ logical freezing. For unselected rows/columns, we set their gradients to zero instead of pruning, ensuring the overall model architecture remains unchanged.
>
> As described in Algorithm 1 of Section 3.1, for the Adam optimizer, before the update step, we mask the gradients based on the selection:
> $$
> \hat{g}_t=\nabla \mathcal{L}_R(W_t) \odot M,
> $$
> where gradients for the unselected rows/columns are zeroed, ensuring no parameter updates occur for them during backpropagation.
>
> > Reported accuracy improvements are modest, focusing more on efficiency than on representational gains.
>
> We acknowledge that the primary contribution of GRFT lies in efficiency rather than significant accuracy improvements. Nevertheless, GRFT achieves accuracy comparable to the main baseline, GPS, while offering substantial storage efficiency. This efficiency advantage is crucial for future optimization and practical applications. For instance, on LLaMA-3 1B, GPS fails to run, whereas GRFT operates normally.
>
> | Dataset | CoLA   | MRPC   | RTE    | Mean Acc. | Params.(%) |
> | ------- | ------ | ------ | ------ | --------- | ---------- |
> | Full    | 0.8428 | 0.8603 | 0.8087 | 0.8373    | 100.00     |
> | LoRA    | 0.8562 | 0.8554 | 0.8159 | 0.8425    | 0.19       |
> | GRFT    | 0.8495 | 0.8554 | 0.8484 | 0.8511    | 0.08       |
>
> The table above presents results for several small GLUE tasks, demonstrating GRFT's versatility in multimodal tasks and large model applications.
>
> > Performance appears somewhat sensitive to hyperparameters such as the number of selected rows and the regularization coefficient , which may affect reproducibility
>
> Thank you for highlighting this issue. We have conducted ablation studies to address this, which are shown in Figure 4 of the original paper. Our experiments reveal that performance exhibits a peak at a certain value for the top-k row/column selection, which is consistent with the characteristics of the original GPS method.
>
> Regarding the regularization coefficient λ, its impact on the results is minimal, with accuracy fluctuations on the order of fractions of a percent for values between $10^{-6}$ and $10^{-4} $. Values of λ within this range produce consistent and strong performance. On the other hand, using excessively large values of λ leads to an overemphasis on maintaining the original model's direction, which weakens the effectiveness of the training. This behavior is common across most regularization techniques.
>
> > No open-source code is provided to confirm claimed memory and runtime benefits.
>
> We will make the complete codebase publicly available in the final version of the paper. This will include the implementation for calculating the GRFT row/column masks and gradient masking.
>
> **Final Remarks**：Thank you again for the insightful comments. We believe the revisions, clarifications, and additional evidence provided above adequately address your concerns. If the responses have resolved the issues you raised, we would greatly appreciate your consideration in raising the score.

---

### Meta-Review · Area_Chair_AbFA · 2026-01-05

**Summary:**

Reviewers broadly agree the paper is technically sound and practically motivated, offering a structured alternative to GPS by selecting whole rows/columns and adding regularization. The decision-relevant concerns were (i) limited conceptual novelty (seen as an incremental extension of GPS), (ii) incomplete/controversial benchmarking against stronger PEFT baselines and limited GLUE coverage, and (iii) insufficient implementation/reproducibility evidence. While the rebuttal improves clarity and partially addresses benchmarking issues, the remaining concerns, especially the unresolved “incremental contribution” framing plus contested/limited empirical coverage and reproducibility, are material and leave the paper below a confident accept threshold.

**Reviewer Concerns:**

Addressed by rebuttal/discussion:

Clarified the intended novelty claim (structured selection vs. element-wise sparsity) and responded to missing implementation details.

Engaged with baseline-comparison issues (incl. VTAB averaging/protocol discussion) and corrected at least one reporting issue.

Still outstanding:

The core “incremental vs. new paradigm” concern remains unaddressed for multiple reviewers.

Benchmark coverage remains a key weakness (stronger vision PEFT baselines; only a subset of GLUE), and one reviewer explicitly maintained their rejection after the rebuttal.

Reproducibility concerns (hyperparameter sensitivity and the lack of released code during review) remain unresolved.

**Reviewer Scores:**

vGHM (6): likely unchanged (rebuttal helps clarity/reproducibility but does not fundamentally change novelty assessment).

KeGi (2): unchanged; explicitly maintained the original score after discussion.

Other reviewers: likely unchanged.

---

### Decision · Program_Chairs · 2026-01-26

Reject